# RAS and PP2A activities converge on epigenetic gene regulation

Anna Aakula[1,*], Mukund Sharma[1,*], Francesco Tabaro[2], Reetta Nätkin[2], Jesse Kamila[1], Henrik Honkanen[1], Matthieu Schapira[3,4], Cheryl Arrowsmith[3,5,6], Matti Nykter[2,7], Jukka Westermarck[1,8,9]

**RAS-mediated human cell transformation requires inhibition of the tumor suppressor protein phosphatase 2A (PP2A). However, the phosphoprotein targets and cellular processes in which RAS and PP2A activities converge in human cancers have not been systematically analyzed. Here, we discover that phosphosites co-regulated by RAS and PP2A are enriched on proteins involved in epigenetic gene regulation. As examples, RAS and PP2A co-regulate the same phosphorylation sites on HDAC1/2, KDM1A, MTA1/2, RNF168, and TP53BP1. We validate RAS- and PP2A-elicited regulation of HDAC1/2 chromatin recruitment, of RNF168-TP53BP1 interaction, and of gene expression. Consistent with their known synergistic effects in cancer, RAS activation and PP2A inhibition resulted in epigenetic reporter derepression and activation of oncogenic transcription. Transcriptional derepression by PP2A inhibition was associated with an increase in euchromatin and a decrease in global DNA methylation. Collectively, the results indicate that epigenetic protein complexes constitute a significant point of convergence for RAS hyperactivity and PP2A inhibition in cancer. Furthermore, the work provides an important resource for future studies focusing on phosphoregulation of epigenetic gene regulation in cancer and in other RAS/PP2A-regulated cellular processes.**

## Introduction

*RAS* genes (*HRAS*, *KRAS*, and *NRAS*) comprise the most frequently mutated oncogene family in human cancer, accounting for 3.5 million new cases yearly, worldwide (Prior et al, 2020). RAS-mediated human cell transformation is preceded by cell immortalization, which is caused by loss of tumor suppressors such as TP53, RB1, or CDKN2 (Minna et al, 2002). Hyperactivation of RAS signaling is, however, not alone sufficient for malignant transformation of immortalized human cells, but requires simultaneous inhibition of the phosphatase activity of the tumor suppressor protein phosphatase 2A (PP2A) (Yu et al, 2001; Hahn et al, 2002; Rangarajan et al, 2004; Sablina et al, 2010; Sato et al, 2013; Tian et al, 2018). PP2A inhibition and *RAS* mutations also significantly synergize in predicting poor overall survival of cancer patients across TCGA pan-cancer data (Kauko et al, 2015). Moreover, reactivation of the tumor suppressor activity of PP2A efficiently inhibits RAS-driven tumorigenesis (Saddoughi et al, 2013; Liu et al, 2015) and synergizes with pharmaceutical targeting of the RAS downstream effector MEK (Kauko et al, 2018). Thus, understanding the mechanistic basis of the synergism between PP2A inhibition and RAS activity could provide novel opportunities for targeting RAS-dependent cancers.

PP2A comprises a family of trimeric protein complexes that counter-balance kinase-mediated phosphorylation throughout cell signaling networks (Fowle et al, 2019). The PP2A trimers are composed of a scaffolding PP2A-A subunit, a catalytic C subunit, and one of the alternative substrate-determining B subunits. In about 10% of human cancers, PP2A is inhibited by genomic mutations, but the most prevalent mechanism for PP2A inhibition in cancer is overexpression of one of the numerous oncogenic PP2A inhibitor proteins such as CIP2A, PME-1, or SET (Kauko & Westermarck, 2018) (Fig 1A). Several downstream effectors/kinases of RAS are identified as PP2A targets, but it is not yet clear what the cancer-relevant cellular processes are in which RAS and PP2A activities converge. Indeed, PP2A has been shown to regulate, for example, RAF-MEK-ERK and PI3K-AKT pathways (Sablina et al, 2010; Kauko & Westermarck, 2018; Fowle et al, 2019), but whether there are processes beyond kinase signaling that are relevant to RAS/PP2A co-operation in cancer is poorly understood. Interestingly, a recent phosphoproteome analysis revealed that PP2A inhibition and RAS activity regulate highly overlapping phosphoproteins (Kauko et al, 2015). However, the functional relevance of these findings has not been studied yet.

[1]Turku Bioscience Centre, University of Turku and Åbo Akademi University, Turku, Finland    [2]Laboratory of Computational Biology, Faculty of Medicine and Health Technology, Tampere University and Tays Cancer Centre, Tampere, Finland    [3]Structural Genomics Consortium, University of Toronto, Toronto, Canada    [4]Department of Pharmacology and Toxicology, University of Toronto, Toronto, Canada    [5]Princess Margaret Cancer Centre, University Health Network, Toronto, Canada    [6]Department of Medical Biophysics, University of Toronto, Toronto, Canada    [7]Foundation for the Finnish Cancer Institute, Helsinki, Finland    [8]Institute of Biomedicine, University of Turku, Turku, Finland    [9]InFLAMES Research Flagship Center, University of Turku, Turku, Finland

Correspondence: jukwes@utu.fi
*Anna Aakula and Mukund Sharma contributed equally to this work

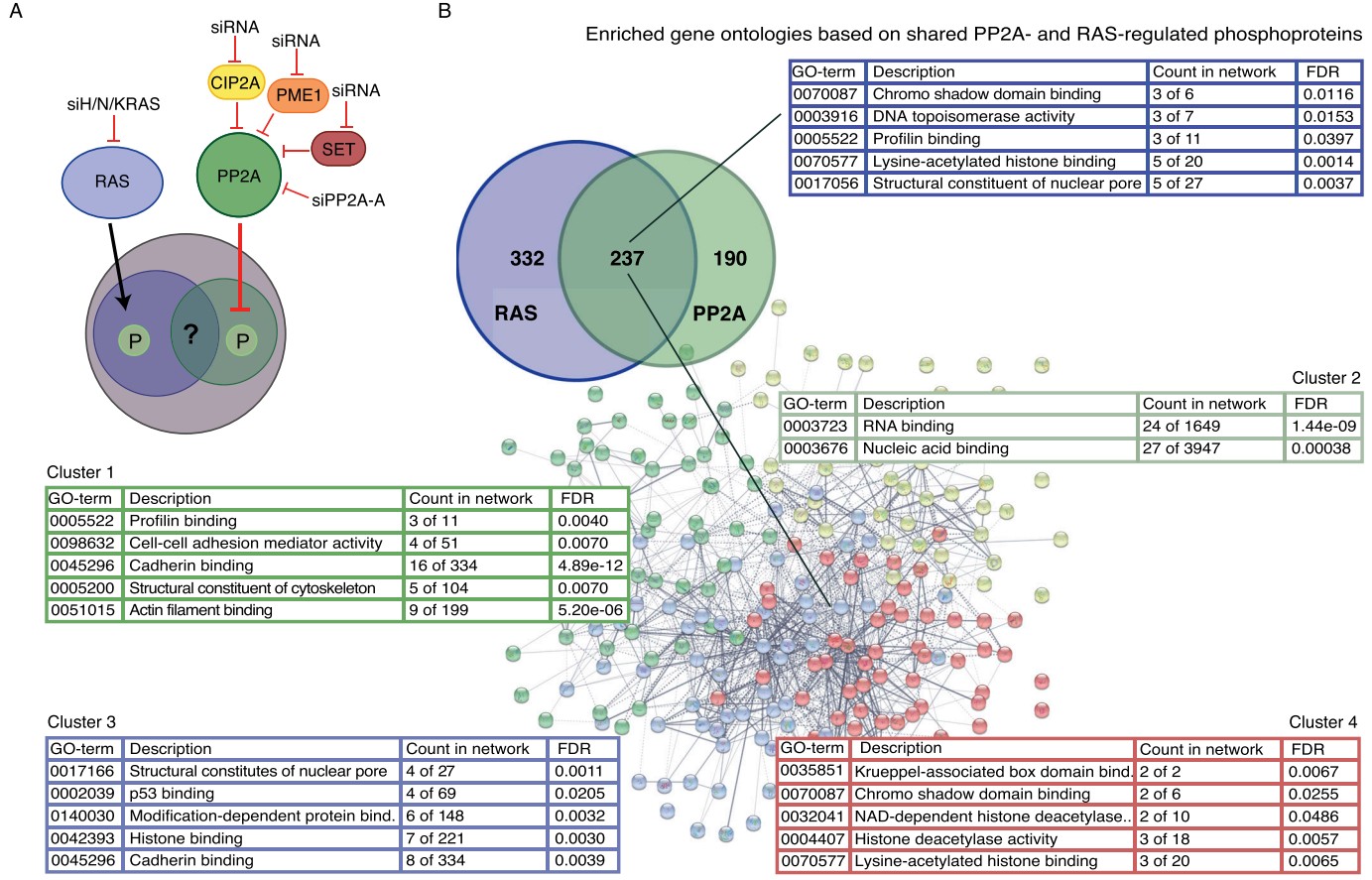

**Figure 1. Enriched gene ontologies based on shared RAS- and PP2A-regulated phosphoproteins.**
**(A)** Schematic presentation of the conducted phosphoproteomics setup, using siRNAs for RAS (H/K/N), PP2A-A, and PP2A inhibitory proteins (CIP2A, PME-1, and SET). **(B)** Figure shows the identified overlap between RAS- and PP2A-regulated phosphoprotein targets, and the enriched gene ontologies (GO terms) related to these. Term PP2A covers targets regulated by siRNA-mediated inhibition of PP2A-A, CIP2A, PME-1, and SET.

Gene expression in multicellular organisms is regulated through various epigenetic processes involving, for example, insertion or removal of chemical tags on the nucleotides and histones (Miller & Grant, 2013). Importantly, both epigenetic gene silencing of tumor suppressors and increased transcription of oncogenic genes contribute to cancer initiation, progression, and therapy resistance (Baylin & Jones, 2016; Quagliano et al, 2020). DNA methylation is the best characterized epigenetic mechanism mediating gene repression. On the contrary, mechanisms that impact nucleosomes via covalent histone modifications leading to open chromatin state (euchromatin) are well established for oncogenic transcription. Strategies to impact epigenetic gene regulation might therefore be useful in cancer prevention and therapy (Cheng et al, 2019; Quagliano et al, 2020).

Although the mechanism of action of epigenetic proteins is extensively studied, and loss-of-function approaches demonstrated that many have a role in cancer (Laugesen & Helin, 2014; Baylin & Jones, 2016), generally very little is known about how phosphorylation-dependent oncogenic signaling regulates their activities (Trevino et al, 2015). One of the major epigenetic complexes involved in cancer is the nucleosome remodeling and deacetylase (NuRD) complex, which functions by two different enzymatic activities: the ATP-dependent nucleosome remodeling through CHD3/4/5 and deacetylation of histone tails through HDAC1/2 (Laugesen & Helin, 2014). HDAC1/2 is one of the rare epigenetic proteins known to be subject to phosphoregulation by kinase/PP2A balance (Bahl & Seto, 2021). However, beyond HDAC1/2, (de)phosphoregulation of the other members of the NuRD complex is poorly understood. Another example of cancer-relevant chromatin remodeling mechanisms is co-operation between RNF168 and TP53BP1, both related to epigenetic gene regulation and DNA damage response (Stewart, 2009; Bohgaki et al, 2013). Based on database information, both proteins are under active phosphoregulation in cancer cells, but currently, there are no indications for the role of RAS or phosphatases in general on their phosphoregulation.

Here, we have addressed the open question of convergence of RAS- and PP2A-mediated phosphoregulation in cancer using previously published phosphoproteome datasets in which RAS proteins and PP2A complexes were targeted by siRNAs (Kauko et al, 2015, 2020) (Fig 1A). The results demonstrate that epigenetic gene regulation is enriched among the cellular processes co-regulated by RAS- and PP2A-mediated phosphorylation. This is due to numerous, but previously unidentified, RAS- and PP2A-regulated

phosphosites in epigenetic proteins implicated in cancer. Functionally, we validate the role of both RAS activity and PP2A inhibition in oncogenic transcription and demonstrate function for PP2A in regulation of DNA methylation and chromatin remodeling. Collectively, the data unveil a previously unknown contribution for PP2A and RAS in phosphoregulation of epigenetic complexes and in oncogenic transcription. The results also provide a rich resource for future investigations of importance of the identified RAS/PP2A-targeted phosphosites in RAS-driven cancers.

# Results

### Systematic analysis of phosphosites co-regulated by RAS and PP2A

To comprehensively map the phosphoproteins co-regulated by PP2A and RAS, we combined data from two recent phosphoproteome studies, in which either all three forms of RAS (HRAS, KRAS, and NRAS) (Kauko et al, 2015), or the PP2A scaffold protein PP2A-A, or the PP2A inhibitor proteins CIP2A, PME-1, and SET (Kauko et al, 2020) were depleted by siRNAs (Fig 1A and Table S1). Previous analysis from these data revealed a role of PP2A in regulating the number of cellular processes not directly related to established PP2A targets such as RAF-MEK-ERK and PI3K-AKT pathways or MYC, but rather to nuclear envelope remodeling, RNA splicing, and DNA damage signaling (Kauko et al, 2015, 2020). However, the convergence between PP2A and RAS phosphotargets has not been addressed thus far.

To accomplish this, we combined the phosphosites that were dephosphorylated in the cells in which either RAS or the PP2A inhibitory proteins (CIP2A, PME-1, and SET) were inhibited, whereas for PP2A-A, we included the phosphosites that had increased phosphorylation (see the Materials and Methods section for data filtering). The resulting RAS/PP2A phosphoproteome consisted of 1,518 unique phosphosites in 749 proteins. RAS inhibition resulted in dephosphorylation of 725 phosphosites in 427 proteins, whereas the corresponding values for PP2A-A and the PP2A inhibitor proteins were 274/195 and 875/441, respectively (Table S2). As a clear indication of convergence of RAS and PP2A activities on phosphoproteome regulation, altogether 270 distinct phosphorylation sites on 237 proteins were found to be co-regulated by both RAS and PP2A targeting (Figs 1B and S1). Interestingly, when assessing the overlap of the regulated phosphosites between RAS and PP2A modulations, sites dephosphorylated by RAS inhibition overlapped more frequently with sites dephosphorylated by PP2A inhibitor protein inhibition, than with sites regulated by PP2A inhibition (Fig S1A and B). This can be explained by the notion that both RAS inhibition and PP2A reactivation inhibit phosphorylation of sites that are constitutively phosphorylated in cancer cells, and based on the recent model that most of the cellular phosphosites are exclusively dominated by either phosphatase activation or inhibition (Kauko et al, 2020). The overlap between RAS and the PP2A inhibitor protein SET was particularly notable (Fig S1A and B). This could explain very potent antitumor effects of SET inhibition in RAS-driven tumorigenesis (Saddoughi et al, 2013; Liu et al, 2015). In

addition to 237 proteins in which at least one phosphorylation site was co-regulated by both RAS and PP2A (Fig 1B), RAS and PP2A co-regulated phosphorylation of 57 overlapping proteins, but in these proteins, the RAS- and PP2A-regulated sites were not identical (Table S2). Collectively, these analyses demonstrate a clear convergence of RAS- and PP2A-mediated phosphoregulation, both at the level of individual phosphorylation sites and at the level of proteins.

To identify cellular processes that would be governed by the convergence of RAS- and PP2A-mediated phosphoregulation, we analyzed enriched gene ontologies (GOs) based on the 237 proteins in which there was at least one phosphosite regulated by both RAS and PP2A (Fig 1B). The STRING database (Szklarczyk et al, 2021) analysis revealed clear enrichment of GOs related to epigenetic and transcriptional gene regulation (Fig 1B). To increase the resolution of the analysis, the shared RAS/PP2A phosphotargets were divided into four clusters (Fig 1B). Whereas cluster 1 was mostly associated with cytoskeleton and cell adhesion, and cluster 2 with nucleic acid binding, clusters 3 and, especially, 4 revealed a significant association of the target proteins with histone modifications and chromatin remodeling (Fig 1B). Although recent data validate the critical role of PP2A in regulating transcriptional elongation (Huang et al, 2020; Vervoort et al, 2021), and epigenetic gene regulation has an important role in RAS-mediated oncogenesis (Vaz et al, 2017), the role of PP2A and RAS in phosphorylation-dependent regulation of epigenetic complexes is very poorly understood. Based on these notions, we focused our downstream analysis on RAS and PP2A convergence on epigenetic gene regulation and transcription.

### PP2A-mediated phosphorylation and RAS-mediated phosphorylation converge on epigenetic complexes

Interestingly, many epigenetic RAS/PP2A phosphotargets were found to constitute protein complexes with each other (Fig 2A). The most apparent examples were the NuRD, DNMT1, and DOT1L complexes (Fig 2A). We hypothesized that RAS/PP2A signaling can potentially regulate DNA methylation, histone methylation, and histone deacetylation via phosphorylation of these complexes (Trevino et al, 2015) (Fig 2A). Consistent with the convergence model, most protein members of these epigenetic complexes were regulated by both RAS and PP2A (Fig 2A), both at the level of individual phosphosites and at the level of proteins (Figs 2B and S2; for all RAS/PP2A-regulated epigenetic target proteins). Naturally, some epigenetic factors were also found to be regulated by either RAS or PP2A only (Figs 2B and S2). In the case of PP2A-regulated targets, the evidence for direct PP2A-mediated dephosphorylation was strengthened by the identification of putative LxxIxE PP2A B-subunit B56 binding motifs (Hertz et al, 2016) in many of these proteins (Table S3). Based on previous evidence that B56 binding motif–containing proteins can act as scaffolds for the recruitment of the other complex proteins for PP2A-mediated dephosphorylation (Hertz et al, 2016), most of the individual PP2A target proteins from our data could become accessible for PP2A-mediated phosphoregulation. Examples of B56 binding motifs on two of the proteins, HDAC1 and SMARCA4, are shown in Fig 2C. Protein–protein interaction between the B56α and HDAC1 was confirmed by reciprocal co-immunoprecipitation analysis (Fig S3A).

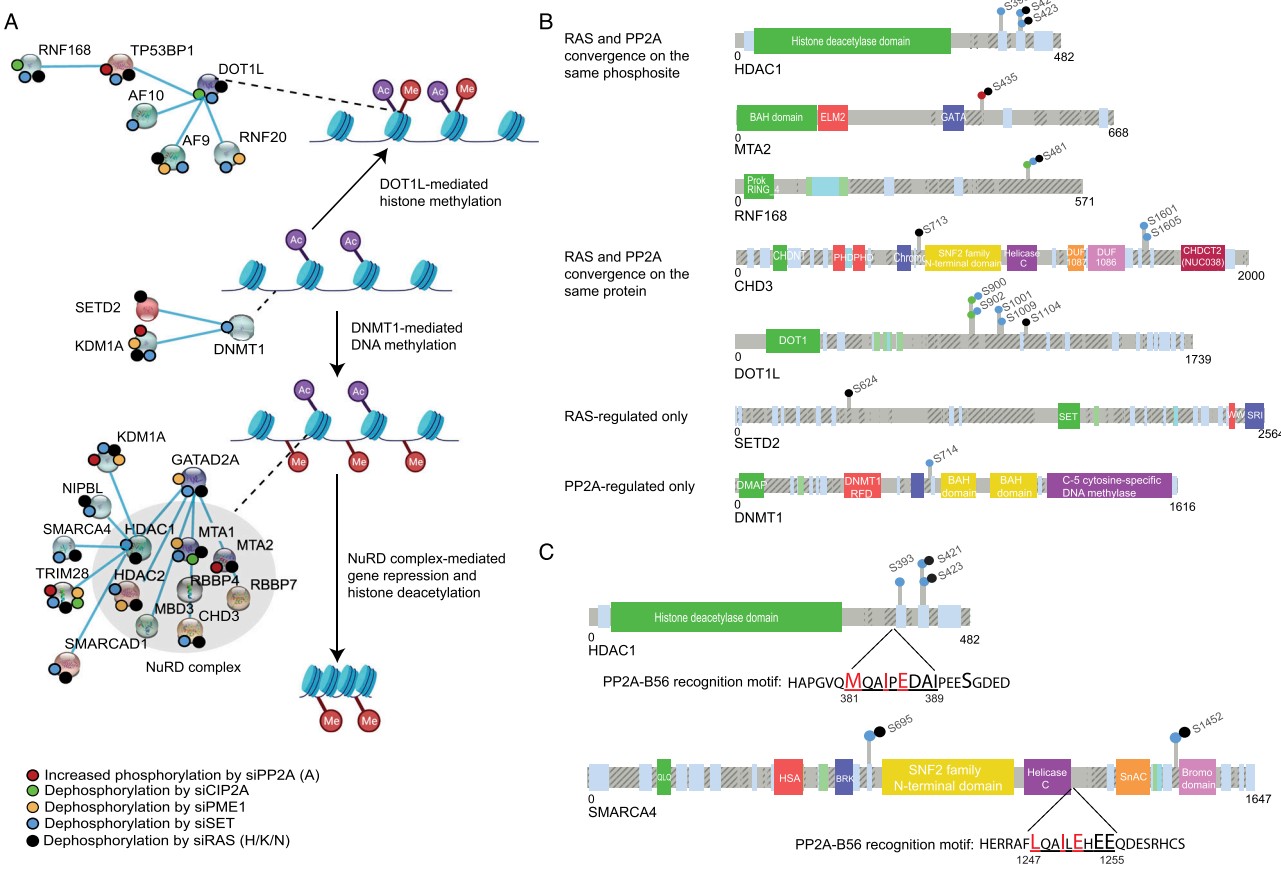

**Figure 2. PP2A-mediated phosphorylation and RAS-mediated phosphorylation converge on epigenetic complexes.**
**(A)** Schematic presents some of the PP2A- and RAS-regulated proteins involved in opening and closing of the chromatin. **(B)** Small colored dots on each of the proteins indicate the identified change in phosphorylation in mass spectrometry upon siRNA treatments of either PP2A-A, its inhibitory proteins (CIP2A, PME-1, or SET), or RAS (H/K/N) (B). **(A)** Schematic presentation of selected PP2A- and RAS-regulated proteins indicating the location of the RAS/PP2A-regulated phosphosites from (A). Similar information regarding the other target proteins is presented in Fig S2. **(C)** Schematic presentation highlighting the PP2A-B56 recognition motif in indicated RAS/PP2A target phosphoproteins. The amino acids indicated with red color are predicted to be essential for B56 binding.

We further characterized the oncogenic potential of selected PP2A/RAS-regulated epigenetic target proteins. To this end, we targeted the selected proteins with five siRNAs per gene, in three KRAS-mutant non–small-cell lung cancer cell lines, A549, H358, and H460 (Fig S3B). In parallel, we tested the cell viability impact of small-molecule targeting of DNMT1, bromodomain proteins, and HDAC1/2 in HCT116 cells (Fig S3C). Collectively, the results show a role of most of the epigenetic RAS/PP2A target proteins in promoting cancer cell viability. These data reveal convergence of PP2A- and RAS-mediated phosphoregulation on cancer-relevant epigenetic protein complexes.

## Impact of PP2A- and RAS-regulated phosphosites on selected epigenetic proteins

Based on the phosphoproteome data (Fig 2), we validated the impact of PP2A- and RAS-regulated phosphosites on selected epigenetic proteins.

Structural evaluation of the phosphorylation sites on selected RAS/PP2A targets provided clues about their potential functional importance. First, PP2A-regulated serine 714 on DNMT1 is located at the base of the loop that must relocate for DNA binding (Fig 3A). On the contrary, using the RNF168 paralog, RNF169–histone 2 complex as the model structure (PDB), the RAS/PP2A-regulated RNF168 serine 481 (S481) was found adjacent to arginine 466 that is critical for histone binding of RNF169 (Fig 3B). Based on our siRNA screening results, RNF168 inhibition resulted in significant inhibition of cell viability across all three tested KRAS-mutant lung cancer cell lines (Fig S3). To probe for potential functional relevance of S481 phosphorylation, we tested the impact of a non-phosphorylatable S481A mutation on interaction of RNF168 with its target protein TP53BP1 (Bohgaki et al, 2013). The co-immunoprecipitation analysis confirmed protein–protein interaction between WT RNF168 and TP53BP1, and this interaction was consistently enhanced when the S481A mutant of RNF168 was used (Fig 3C and D). Equal transgene expression of RNF168 WT and mutant construct was confirmed by qRT-PCR (Fig S3D).

CHD3 instead is a member of the NuRD complex, and it also supports viability of KRAS-mutant lung cancer cell lines (Fig S3). Based on the PhosphoSitePlus database (www.phosphosite.org), CHD3 is phosphorylated on 54 distinct serines or threonines, but there are no reports about the functional relevance of any of these

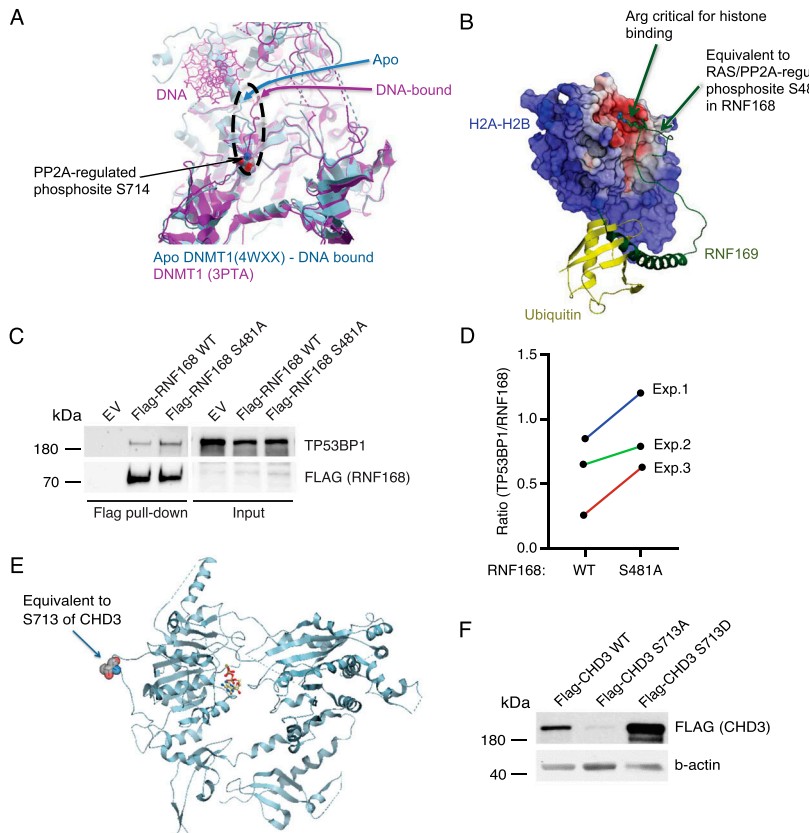

**Figure 3. Functional relevance of RAS/PP2A-regulated phosphosites on epigenetic proteins.**
**(A)** Structural modeling of the PP2A-regulated phosphosite S714 on DNMT1. **(B)** Structural modeling of RAS/PP2A-regulated phosphosite S481 on RNF168 using RNF169–histone 2A/2B co-crystal structure as a model. **(C)** Co-immunoprecipitation analysis of the impact of RNF168 serine 481 (S481) phosphorylation on protein interaction with TP53BP1. **(D)** H460 cells were transfected with either empty expression vector (EV) or indicated RNF168 plasmids, and FLAG-trap pull-down assay was performed after 48 h (n = 3) (D). **(C)** Quantification of the ratio between TP53BP1 and RNF168 in pull-down samples from three independent experiments as shown in (C). **(E)** CHD3 protein and the location of the identified RAS-regulated serine 713. **(F)** Western blot analysis of the effect of the S713A and S713D mutations on protein stability upon CHD3 overexpression in A549 cells.

phosphosites. In our data, RAS inhibition resulted in dephosphorylation of S713, whereas PP2A reactivation by SET inhibition caused dephosphorylation of S1601 and S1605 (Fig 2B and Table S1). Using yeast CHD1 crystal structure (PDB: 3MWY) as a model, the RAS target site S713 was located on the unstructured region in the vicinity of the nucleotide binding mediating region between amino acids 761 and 768 in human CHD3 protein (Fig 3E). This structural organization was supported by AlphaFold analysis of human CHD3 (Fig S4A). Notably, S713 phosphorylation in KRAS-mutant A549 lung cancer cells seems to be critical for the stability of CHD3 protein, as S713A mutation dramatically inhibited CHD3 protein expression, whereas the phosphorylation mimicking mutation S173D resulted in increased protein expression (Fig 3F).

Collectively, these data provide important indications about the functional relevance of RAS- and PP2A–co-regulated phosphosites in epigenetic proteins. However, understanding of the functional role of each identified phosphorylation site reported here will require extensive validation experiments outside the scope of this resource article.

## Impact of PP2A and RAS on NuRD complex chromatin recruitment

Next, we evaluated the potential impact of PP2A/RAS activities on NuRD complex chromatin recruitment by following the intranuclear distribution of HDAC1/2 as central components of the NuRD complex. The NuRD complex was chosen as it showed very high

level of PP2A/RAS-mediated phosphorylation regulation, and as HDAC1 and HDAC2 had a conserved B56 binding motif (Fig 2C and Table S3), and HDAC1 interaction with PP2A-B56 was validated by co-immunoprecipitation (Fig S3A).

Notably, PP2A activation, either by siPME-1 or by three pharmacological activators DBK1154, DT061, and FTY720 (Vainonen et al, 2021), all enhanced the chromatin recruitment of HDAC1/2 (Fig 4A–D). Consistent with the prominent impact of RAS on NuRD complex phosphorylation (Fig 2A), RAS inhibition also resulted in increased chromatin retention of HDAC1 and HDAC2 (Fig 4E and F). To link the HDAC regulation by PP2A inhibition and RAS activity to their co-operative roles in human cell transformation (Yu et al, 2001; Hahn et al, 2002; Rangarajan et al, 2004; Sablina et al, 2010; Sato et al, 2013; Tian et al, 2018), we evaluated chromatin recruitment of HDAC1 from human bronchial epithelial cells (HBECs) transformed by serial introduction of short hairpin p53 (p53-), RAS G12V overexpression (KRAS+), and overexpression of the viral PP2A inhibitor protein small-t (ST). Consistent with the published results (Sato et al, 2013), only the HBECs with both RAS activation and PP2A inhibition were able to grow on soft agar as a measure of cellular transformation (Fig 4G and H). Interestingly, p53 inhibition in HBECs resulted in robust chromatin recruitment of HDAC1 and HDAC2 (Fig 4I and J). However, the chromatin recruitment was reversed by RAS activation, and this was further stabilized by ST-elicited PP2A inhibition (Fig 4I and J). These results directly link human cell transformation requirements to epigenetic gene regulation by HDAC1/2.

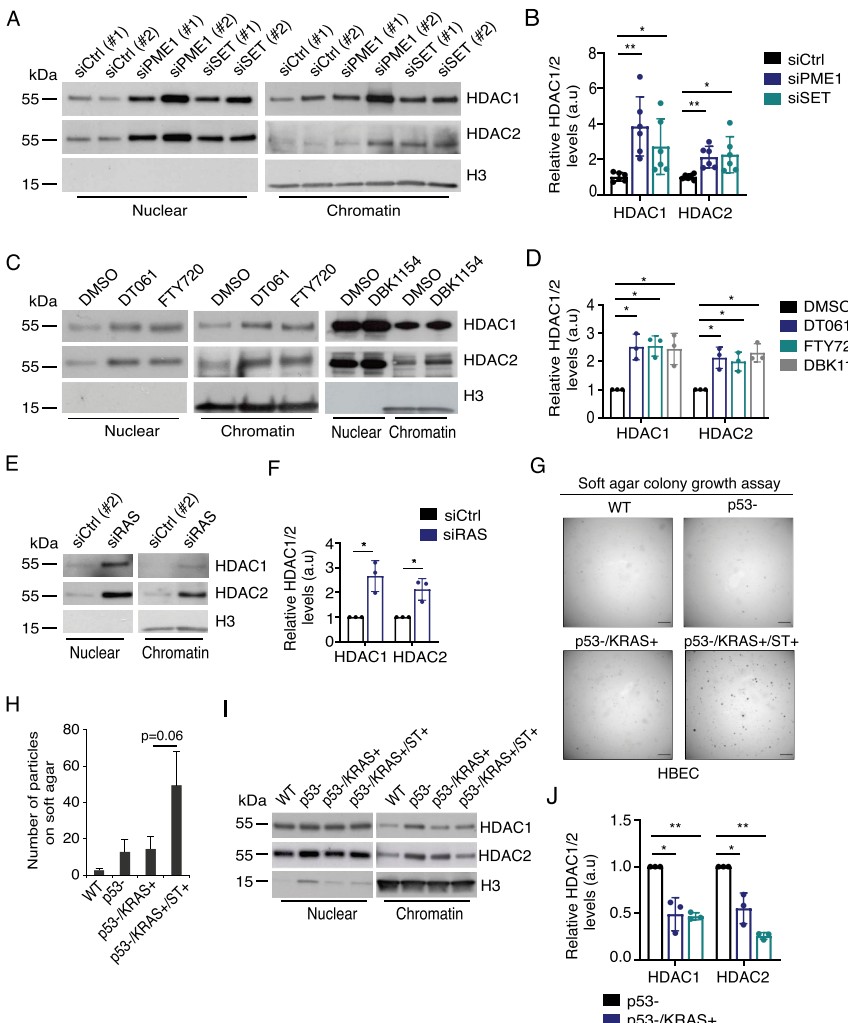

**Figure 4. RAS activation and PP2A inhibition inhibit chromatin recruitment of HDAC1/2 upon human cell transformation.**

**(A, B)** Western blot analysis of chromatin recruitment of HDAC1 and HDAC2 upon siRNA-mediated inhibition of indicated PP2A inhibitory proteins in H460 cells, (B). Quantification of results from (A) (n = 3). **(C)** Western blot showing the impact of pharmacological PP2A activation (DT061, DBK1154, and FTY720) on HDAC1/2 recruitment to chromatin in H460 cells. Each drug was used at 10 μM concentration for 48 h. **()** Quantification of (C) (n = 3). **(E)** Western blot analysis of the impact of RAS inhibition on HDAC1/2 chromatin recruitment in H460 cells. **(E, F)** Quantification of chromatin recruitment upon siRAS from WB in (E) (n = 3). **(G)** Human bronchial epithelial cells were step-wise–transformed by indicated genetic changes: p53- (TP53shRNA), KRAS+ (KRASG12V overexpression), and ST+ (SV40 small-t antigen overexpression to inhibit PP2A). Anchorage-independent growth capacity as an indication of transformation status was subsequently tested on soft agar assay. Particles smaller than 200 μm² were not considered colonies. Shown are representative images depicting the colonies. Scale bar: 1,650 μm. **(G, H)** Quantification of the number of colonies from (G), showing the mean ± SEM from three independent experiments. **(I)** Chromatin recruitment of HDAC1/2 in the step-wise–transformed human bronchial epithelial cells. **(I, J)** Quantification of (I) (n = 3). **(B, D, F, J)** Quantitation refers to the relative amount of HDAC proteins in the chromatin fraction as compared to the control condition set as 1. Shown is the mean + SD from three experiments. * $P < 0.05$ and ** $P < 0.01$.

To provide translational relevance, we further asked whether co-operative inhibition of HDAC1/2 chromatin recruitment by RAS activity and PP2A inhibition impacts cellular sensitivity to the pharmacological HDAC inhibition. To this end, PP2A-A was depleted from KRAS-mutant H460 cells, and the cells were treated with increasing concentrations of the clinical-stage HDAC inhibitor panobinostat (Fig S5A). Alternatively, we tested the potential synergy between panobinostat and PP2A activator DBK1154 (Fig S5B and C). In both settings, we observed that the impact of PP2A on nucleoplasmic/chromatin distribution of HDAC1/2 correlated with the sensitivity of cells to panobinostat. Decreased chromatin recruitment by PP2A inhibition (Fig 4I and J) correlated with panobinostat resistance, whereas increased chromatin recruitment by PP2A activation (Fig 4A–D) correlated with increased panobinostat sensitivity. Importantly, none of the RAS/PP2A modulations used above impacted HDAC1/2 total protein expression, as studied from the whole-cell extracts (Fig S5D), indicating that the observed differences are due to selective regulation of HDAC1/2 on chromatin fraction.

These results demonstrate that the combined requirements for human cell transformation, that is, RAS activation and PP2A

inhibition (Yu et al, 2001; Hahn et al, 2002; Rangarajan et al, 2004; Sato et al, 2013), result in decreased HDAC1/2 recruitment to chromatin, and that this can be reversed by PP2A reactivation.

**PP2A inhibition and RAS activity derepress transcription**

To address the functional impact of RAS- and PP2A-mediated co-regulation of epigenetic complexes on gene regulation, we employed a previously established HCT116 stable cell line carrying an epigenetically silenced *SFRP1* promoter–GFP reporter (Cui et al, 2014) (Fig 5A). Derepression of the promoter results in GFP expression that can be monitored either by Western blotting or by live-cell IncuCyte analysis (Cui et al, 2014). As a technical control, we verified that treatment of cells with the DNMT1 inhibitor 5-aza-2'-deoxycytidine (decitabine) resulted in increased *SFRP1*-GFP reporter activity (Fig 5B). Notably, the reporter was also responsive to pharmacological inhibition of other epigenetic PP2A/RAS target mechanisms, such as HDACs, BET proteins, and KDM1A (Figs 5B and S5E). This validates suitability of this cell model as a surrogate reporter for PP2A/RAS-mediated regulation of epigenetic gene regulation.

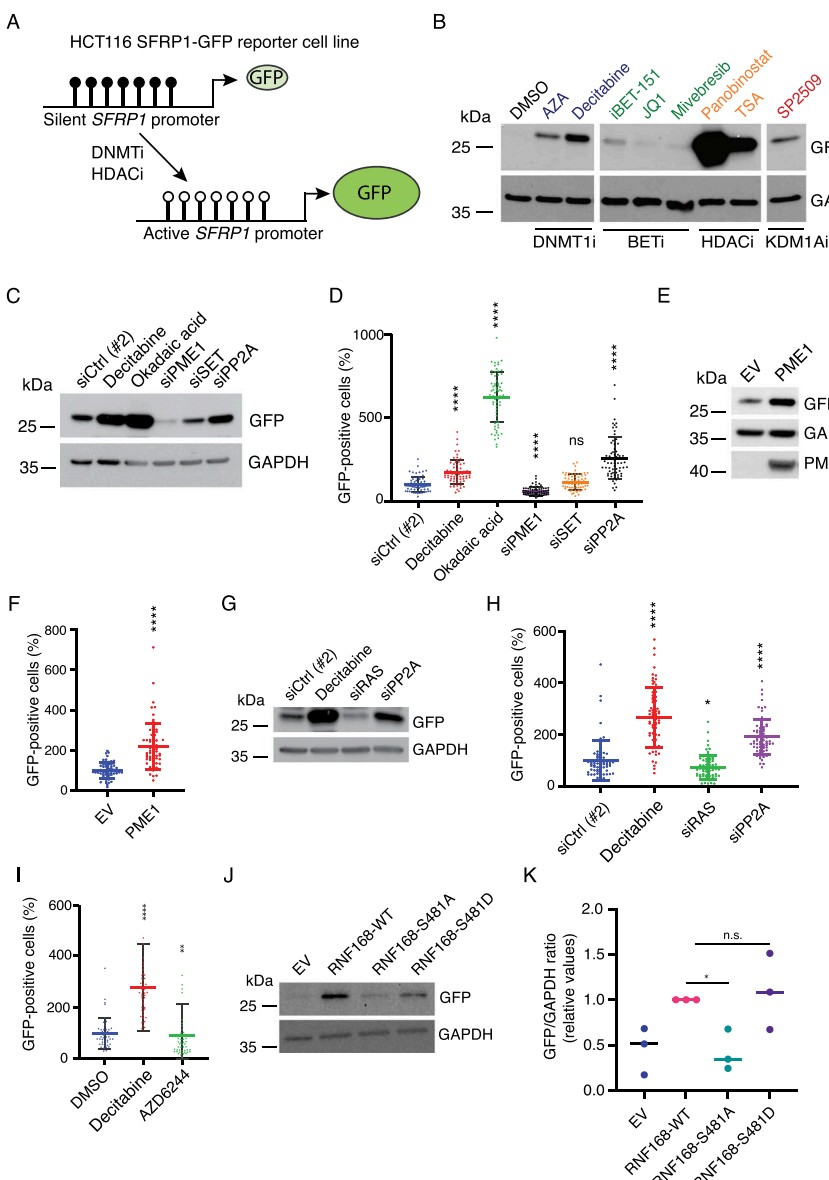

**Figure 5. Opposing roles of PP2A and RAS in transcription from an epigenetically silenced SFRP1 promoter.**
**(A)** Schematic representation of the *SFRP1*-GFP reporter system, in which GFP expression from silenced *SFRP1* promoter can be activated by treatments that derepress gene expression, for example, by inhibition of DNMT1-mediated DNA hypermethylation or by HDAC inhibition. **(B)** Validation of the suitability of the reporter system for detecting gene activation upon pharmacological targeting of PP2A/RAS-regulated epigenetic mechanisms. Shown are the Western blot analyses of HCT116 *SFRP1*-GFP reporter cells after 48 h of treatment with indicated drugs at the following IC50 cell viability concentrations (see Fig S3C): azacitidine, 22.24 μM; decitabine, 10 μM; iBET151, 18.78 μM; JQ1, 5.26 μM; mivebresib, 3.09 μM; panobinostat, 0.278 μM; and TSA, 0.246 μM. **(C)** Western blot analysis of HCT116 *SFRP1*-GFP reporter cells after treatment with decitabine (5 μM), PP2A activation (siPME-1 and siSET for 72 h), or PP2A inhibition (okadaic acid [OA, 25 nM, 12 h] or siPP2A for 72 h) (n = 3). **(B, D)** IncuCyte analysis of fluorescence in HCT116 *SFRP1*-GFP reporter cells after the same treatments as in (B). **(E)** Western blot analysis of reporter cells after the overexpression of the PP2A inhibitor protein PME-1 for 48 h (n = 3). **(F)** Analysis of fluorescence in reporter cells after the overexpression of PME-1. **(G)** Western blot analysis of reporter cells after treatment with decitabine (5 μM, 48 h), or siPP2A and siRAS for 72 h (n = 3). **(F, H)** Analysis of fluorescence in reporter cells in the same conditions as in (F). **(I)** Analysis of fluorescence in reporter cells upon treatment with decitabine or MEK inhibitor AZD6244 (5 μM each) for 72 h as compared to control (DMSO) (n = 5). **(J)** Western blot analysis of reporter cells after the overexpression of either WT, S481A, or S481D mutants of RNF168 for 48 h. **(J, K)** Quantification of the ratio between GFP and GAPDH expression from (J). Shown is the mean from three biological replicates. *P* < 0.05, two-tailed *t* test. **(D, F, H, I, K)** Shown is the mean + SD. *P* < 0.05 and **-**** *P* < 0.01, Mann–Whitney U test.

To test the impact of RAS/PP2A activities on epigenetically repressed *SFRP1* promoter activity, we used the same treatments as were used for generating the phosphoproteome data (Kauko et al, 2015, 2020). Notably, inhibition of PP2A either by siPP2A-A or by chemical serine/threonine phosphatase inhibitor okadaic acid resulted in increased promoter activity measured either by Western blotting or by IncuCyte (Fig 5C and D). Related to the endogenous PP2A inhibitory mechanisms, PME-1 depletion resulted in further repression of *SFRP1*-GFP reporter activity, whereas SET depletion did not have an effect (Fig 5C and D). The role of PME-1–mediated PP2A inhibition in promoting oncogenic transcription was further supported by the increased reporter activity upon transient PME-1 overexpression (Fig 5E and F). On the contrary, RAS inhibition resulted in a significant decrease in *SFRP1*-GFP reporter activity (Fig 5G and H). Downstream of RAS, the effect on reporter activity appears to be at least partly mediated by the MEK-ERK MAPK pathway,

as MEK inhibitor AZD6244 treatment also resulted in significant reporter activity inhibition (Fig 5I). Lastly, to link PP2A/RAS-mediated regulation of identified epigenetic target protein phosphosites to gene regulation, we tested the impact of RNF168 WT and S481 phosphomutants on *SFRP1*-GFP promoter activity. Overexpression of WT RNF168 resulted in transcriptional activation, and consistent with the model that RNF168 S481 is constitutively phosphorylated in cancer cells, this effect was indistinguishable from the effects with the constitutive phosphorylation mimicking S481D mutant (Fig 5J and K). However, the S481A mutant was fully incapable of inducing transcription from the repressed *SFRP1* promoter (Fig 5J and K), despite equal expression of all RNF168 variants at the protein level (Fig S5F).

These data indicate that opposing roles of PP2A and RAS in oncogenesis (Rangarajan et al, 2004; Sato et al, 2013; Zhou et al, 2017) could at least partly be explained by their opposite roles in

gene expression via phosphoregulation of epigenetic target phosphoproteins.

## Transcriptional profiling of PP2A- and RAS-inhibited cells

We analyzed global gene expression patterns upon PP2A or RAS inhibition by RNA sequencing (RNA-seq). Consistent with the results obtained by the *SFRP1*-GFP reporter system, PP2A inhibition predominantly resulted in global gene activation, whereas RAS inhibition predominantly led to gene repression (Fig 6A and B). Notably, gene set enrichment analysis (GSEA) of genes activated by PP2A

inhibition showed significant association with KRAS up-regulation gene signature, indicating a novel transcriptional layer for PP2A-mediated antagonism of RAS signaling (Fig 6C). Other significant cancer-associated pathway signatures up-regulated upon PP2A inhibition included epithelial–mesenchymal transition and mitotic spindle (Fig 6C). Genes down-regulated by PP2A inhibition were instead not enriched to any cellular process, supporting the conclusions that PP2A inhibition conveys its oncogenic effects primarily by gene activation. On the contrary, GO analysis of PP2A-inhibited cells showed enrichment of "positive regulation of intracellular signal transduction," indicating that in addition to its

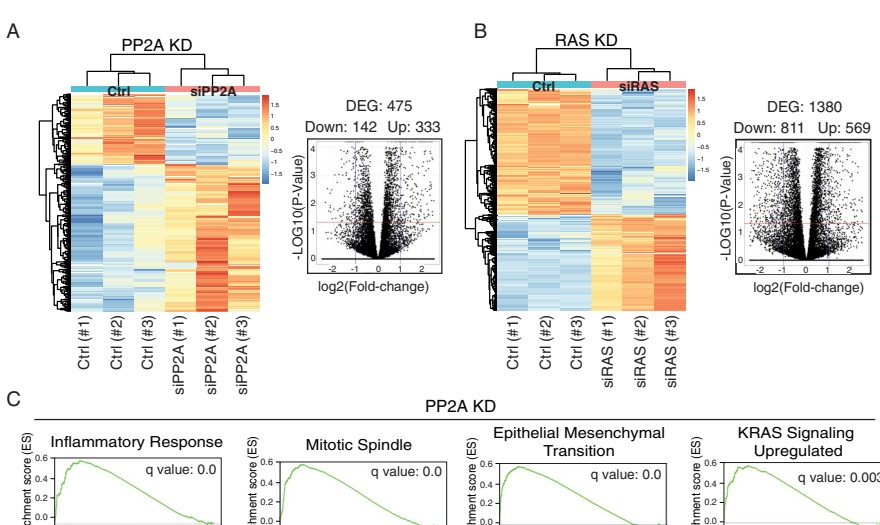

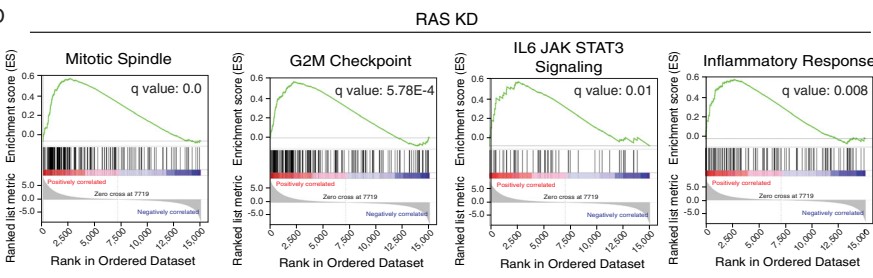

**Figure 6. Convergence of PP2A and RAS activities on oncogenic gene expression in HeLa cells.**
**(A)** Heatmap of RNA-seq analysis of cells after PP2A inhibition and the corresponding volcano plot showing differentially regulated genes. **(B)** Heatmap of RNA-seq analysis of cells after RAS inhibition and the corresponding volcano plot showing differentially regulated genes. **(C)** Enrichment plots from gene set enrichment analysis of PP2A-inhibited genes. **(D)** Gene set enrichment analysis enrichment plots of RAS-inhibited genes. **(E)** Top 10 transcription factors regulated upon PP2A silencing. **(F)** Top 10 transcription factors regulated upon RAS silencing. The overlapping transcription factor elements enriched in both PP2A and RAS targets are bolded.

**E** PP2A KD

| Transcription Factor | Rank | 1st Sample p-value |
|---|---|---|
| **TEAD1** | 1 | 4.23E-28 |
| YAP1 | 2 | 5.23E-23 |
| **AR** | 3 | 4.73E-21 |
| TEAD4 | 4 | 6.89E-20 |
| SMAD3 | 5 | 3.09E-19 |
| **NR3C1** | 6 | 1.45E-18 |
| RELA | 7 | 2.38E-18 |
| HOXB13 | 8 | 3.10E-16 |
| **FOSL2** | 9 | 9.53E-16 |
| LHX2 | 10 | 2.10E-15 |

**F** RAS KD

| Transcription Factor | Rank | 1st Sample p-value |
|---|---|---|
| **NR3C1** | 1 | 9.84E-13 |
| **FOSL2** | 2 | 6.46E-12 |
| HES2 | 3 | 1.18E-11 |
| CEBPB | 4 | 6.61E-11 |
| **TEAD1** | 5 | 7.48E-09 |
| FOS | 6 | 2.71E-07 |
| FOXA1 | 7 | 7.25E-07 |
| **AR** | 8 | 8.19E-07 |
| HNF1B | 9 | 3.08E-06 |
| CALU | 10 | 3.49E-06 |

direct role in protein dephosphorylation, PP2A regulates signaling pathway activities transcriptionally (Fig S6). Other significant GO terms indicated the role of PP2A in regulation of cellular adhesion, migration, and motility, all highly relevant for malignant cancers (Fig S6). GSEA of genes down-regulated by RAS inhibition revealed an overlap with PP2A-regulated GSEA signatures, such as inflammatory response and mitotic spindle (Fig 6C and D). In addition, RAS was found to regulate genes related to IL-6-JAK-STAT signaling and G2M checkpoint (Fig 6D). On the contrary, the most significantly enriched GO biological process upon RAS silencing was "negative regulation of cell proliferation activity" (Fig S6).

Notably, the convergence of RAS and PP2A activities on onco-genic transcription was also apparent via analysis of enrichment of transcription factor binding motifs on differentially regulated genes. TEAD (TEAD1, TEAD4, and YAP) and FOS (FOS and FOSL2) target genes were significantly enriched among those regulated by both PP2A and RAS targeting (Fig 6F, in bold). This is consistent with recent results that PP2A complexes containing striatins stimulate YAP1 activity leading to cellular transformation (Kurppa & Westermarck, 2020). On the contrary, YAP1 drives resistance to KRAS and EGFR inhibition (Shao et al, 2014; Kurppa et al, 2020). Furthermore, KRAS and YAP1 synergistically activate FOS

transcription factors, leading to epithelial–mesenchymal transition (Shao et al, 2014). In addition, AR target genes were also enriched among PP2A- and RAS-regulated targets (Fig 6F, in bold). This is consistent with previous results indicating role of both RAS activity and PP2A inhibition in promoting malignant growth of AR-positive prostate cancers (Weber & Gioeli, 2004; Khanna et al, 2015).

Collectively, the RNA-seq data further support the conclusions that PP2A inhibition drives oncogenic transcription and that PP2A and RAS activities converge on transcriptional regulation of gene expression at various levels.

## DNA methylation and chromatin remodeling effects by PP2A inhibition

Because both the PP2A-regulated phosphoproteome targets (Fig 2A) and increased activity of the methylation-sensitive reporter assay (Fig 5) indicated that PP2A could regulate DNA methylation, we analyzed global DNA methylation in PP2A-inhibited cells by reduced-representation bisulfite sequencing (RRBS). Consistent with the transcriptional derepression (Figs 5 and 6), the siRNA-mediated inhibition of PP2A-A predominantly resulted in DNA demethylation (Fig 7A and B). Of the total 211 differentially

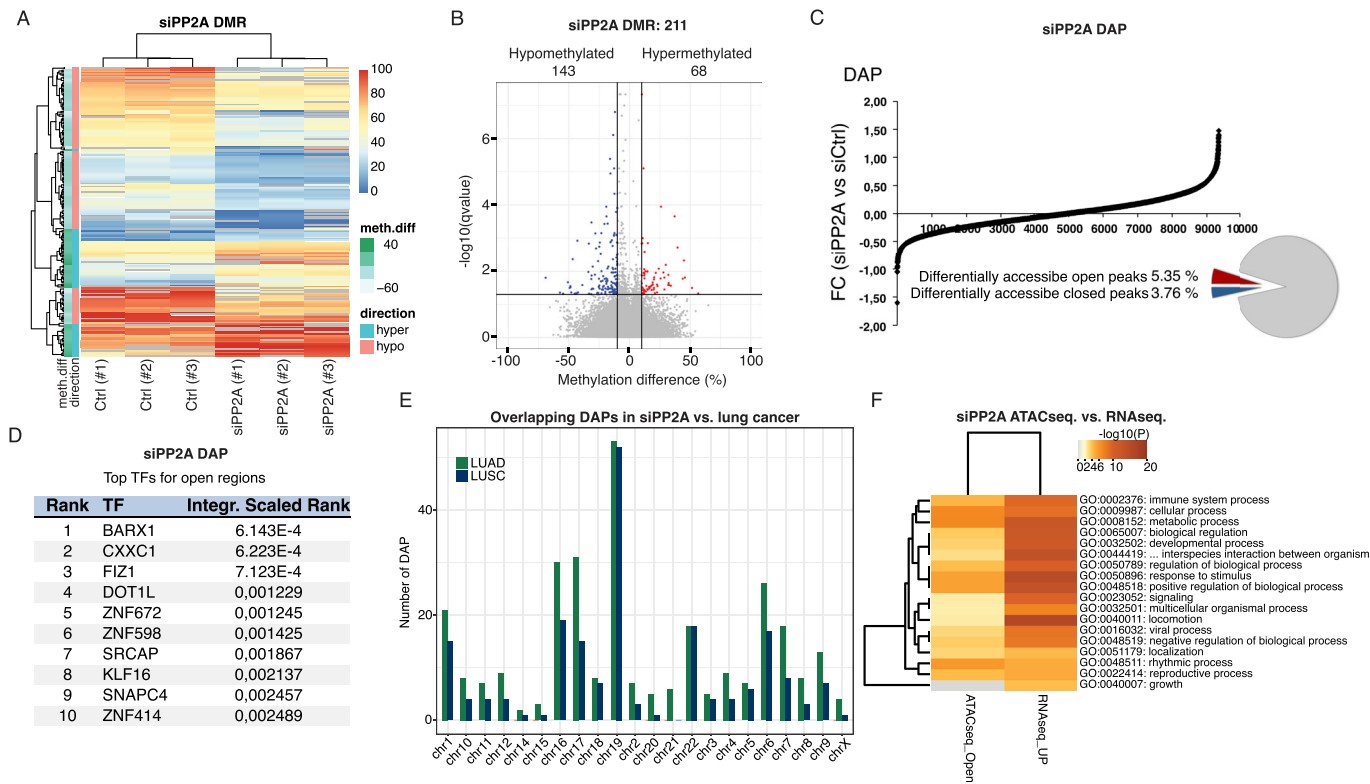

**Figure 7. DNA demethylation and chromatin remodeling upon PP2A inhibition in HeLa cells and overlap of clinical patient sample chromatin landscapes.**
**(A)** Heatmap showing the sample grouping according to CpG methylation levels (red = 100% methylated, blue = 0% methylated) upon PP2A inhibition. **(B)** Volcano plot for differentially methylated regions, the blue dots indicate the significantly regulated hypomethylated regions, whereas the red dots indicate the hypermethylated regions. **(C)** Graph illustrates the total number of open/closed areas that has changed upon PP2A-A knockdown as compared to siCtrl. Of the identified peaks, 5.35% are opened and 3.76% closed (FC ± 0.5) in siPP2A versus siCtrl. **(D)** Transcription factor binding elements enriched in genes in proximity to differentially accessible gene promoters. **(E)** Chromosomal distribution of differentially accessible peaks overlapping between siPP2A-treated HeLa cells and lung cancer clinical patient material. LUAD, lung adenocarcinoma; LUSC, lung squamous cell carcinoma. **(F)** Heatmap of overlap between SiPP2A-A–elicited transcriptional up-regulation (RNAseq_up) or open chromatin region (ATACseq_Open) of genes involved in indicated cellular processes.

methylated regions, 143 regions showed a decrease in methylation marks (hypomethylated), whereas 68 showed an increase in methylation (hypermethylated) (Fig 7B). The occupancy of the differentially regulated methylation marks was lowest at the exons (7%), while almost symmetrically distributed between introns (36%), intergenic areas (30%), and promoter regions (27%) (Fig S7A). Inhibition of PP2A had an overall maximum impact on methylation of chromosome 11 (Fig S7B). On the contrary, when the ratio between hypomethylation and hypermethylation was considered, the highest degree of hypomethylation was seen in X chromosome, which was exclusively hypomethylated, followed by chromosome 13 in which six regions were hypomethylated and only one region was hypermethylated in response to PP2A inhibition (Fig S7C).

To interrogate PP2A function in oncogenic transcription via its impact on DNA methylation, we performed a GO enrichment analysis based on differentially methylated regions and using the Enrichr analysis tool (Kuleshov et al, 2016) (Fig S7D–G). Importantly, overlapping with the GSEA of PP2A-regulated gene expression (Fig 6C), both epithelial-to-mesenchymal transition and several gene sets related to membrane-associated GTPase activity (corresponding to KRAS activity in GSEA) were enriched in PP2A-inhibited cells (Fig S7D–G).

The phosphoproteome targets (Fig 2A) indicated that PP2A may regulate gene expression also by affecting chromatin accessibility. This prompted us to perform Assay for Transposase-Accessible Chromatin using sequencing (ATAC-seq) analysis from cells transfected with control and PP2A A-subunit siRNAs. Supporting our hypothesis, PP2A inhibition induced marked changes in the pattern of open chromatin regions (Figs 7C and S8).

The number of open peaks in PP2A-silenced cells was 9,347, and the general distribution resembled that of ATAC-seq profiles in general (Yan et al, 2020) (Fig S8A and B). The number of differentially accessible peaks (DAPs) upon PP2A silencing was instead 851. The most enriched binding sites for transcriptional regulators associated with genes with open and closed promoter regions are listed in Figs 7D and S8C, respectively. Some of the DAPs in PP2A-inhibited cells were found in proximity to genes involved in chromosome and chromatin binding (e.g., *KMT2A*), indicating another putative layer of regulation of how PP2A inhibition may impact oncogenic transcription.

Next, we were interested in whether the observed PP2A-dependent modulation of chromatin accessibility could have clinical relevance. Therefore, we compared the DAPs from the PP2A-inhibited cells with non–small-cell lung cancer patient ATAC-seq profiles, including lung adenocarcinoma and lung squamous cell carcinoma (Wang et al, 2019). Interestingly, several DAPs identified in the PP2A-inhibited cells were found to overlap with open regions in lung cancer patient samples (Fig 7E) with the highest degree of overlap in promoter regions (Fig S8D). The overlap between DAPs in siPP2A samples, and open chromatin peaks in clinical samples was statistically significantly enriched ($P < 0.05$) in chromosomes 6, 7, and 19 in lung adenocarcinoma, and in chromosomes 6, 18, 19, and 22 in lung squamous cell carcinoma (Table S4). Interestingly, chromosome 19 aberrations are frequently observed in human lung cancers (Wang et al, 2015). By analyzing the nearest genes to the PP2A-overlapping DAPs in chromosome 19 (Table S5), we identified, for example, ferroptosis regulator *GPX* (Doll et al, 2019), strongly implicated in lung cancer progression and metastasis (Zou et al,

2021), or MAPKK MKK7 (MAP2K7), which is an upstream regulator of JNK driving lung tumorigenesis downstream of RAS (Ruiz et al, 2021).

Based on the presented data, PP2A inhibition results in increased transcription, DNA hypomethylation, and open chromatin. All these changes enhance transcription and thus demonstrate concerted action by PP2A inhibition at the level of epigenetic (oncogenic) gene regulation. We further asked whether there were consistent biological processes regulated by PP2A inhibition based on data from all omics levels: RNA-seq, RRBS, and ATAC-seq. Indeed, there was a strong overlap between enriched biological processes based on PP2A-regulated ATAC-seq and RNA-seq data (Fig 7F), and between PP2A-regulated RRBS and RNA-seq data (Fig S9). We also integrated all three levels of epigenetic regulation by PP2A. The overlap across all three technologies was naturally more limited, but interestingly, overlaps of cancer- and RAS/PP2A-relevant processes such as "signaling by RTK" and "kinase signaling" were identified (Fig S9B).

These results validate the hypothesis that PP2A regulates DNA methylation and chromatin accessibility and provide indications for a potential cancer relevance of the newly identified epigenetic function of PP2A. Future studies are needed to functionally link the observed RAS/PP2A epigenetic phosphotargets to these global gene regulation effects.

## Discussion

Epigenetic gene regulation has an established role in cancer initiation and progression (Laugesen & Helin, 2014; Baylin & Jones, 2016; Cheng et al, 2019; Quagliano et al, 2020). Several loss-of-function studies across different species have also demonstrated an integral role of epigenetic gene regulation in signal transduction, development, and malignant progression downstream of RAS proteins (Vaz et al, 2017). However, it has been surprisingly poorly known how RAS impacts phosphorylation of epigenetic proteins, and whether RAS activity toward epigenetic gene regulation is modulated by PP2A-mediated protein dephosphorylation. Here, we provide the first bird's-eye view of the global impact of RAS and PP2A activities on the phosphorylation regulation of epigenetic complexes and their co-operative impact on oncogenic gene regulation. The results indicate that epigenetic protein complexes involved in oncogenic gene expression constitute a significant point of convergence for RAS hyperactivity and PP2A inhibition in cancer. They also indicate that functional interactions between RAS and PP2A in cancer cannot be solely explained by previously implicated kinase and MYC regulation (Yeh et al, 2004; Sablina et al, 2010; Fowle et al, 2019). The results also provide a very rich resource for future interrogation of the impact of the identified phosphosites both in physiological gene regulation and in human cancer development and progression.

About 20 yr ago, inhibition of serine/threonine phosphatase activity of PP2A was established by several studies as a prerequisite for RAS-mediated malignant transformation of human, but not of mouse, cells (Yu et al, 2001; Hahn et al, 2002; Rangarajan et al, 2004). Mechanistically, the requirement of PP2A inhibition for RAS-mediated human cell transformation was attributed to the role

of PP2A as an inhibitor of the activities of several downstream mediators of RAS activity such as MEK/ERK and AKT kinases, or transcription factor MYC (Yeh et al, 2004; Sablina et al, 2010). In this prevailing model, PP2A inhibition is seen merely as a mechanism to boost signal transduction initiated by RAS. However, it has not been previously systematically addressed whether their activities would converge on particular cellular mechanisms or processes. This is a critical unanswered question, as understanding why RAS activity and PP2A inhibition are mutually required for human cell transformation could lead to fundamental novel understanding of the basis of human cancer development. Furthermore, as shown with some examples already in this study, this understanding could facilitate novel therapeutic approaches that target the roots of human malignancies. In this context, PP2A inhibition has been demonstrated to drive resistance of KRAS-mutant cells toward a wide array of kinase inhibitors (Kauko et al, 2018). Furthermore, we and others have shown that PP2A inhibition drives cancer cell resistance to epigenetic therapies (Fig S5A–C) (Shu et al, 2016; Kauko et al, 2020). Thereby, the presented results may provide important clues for understanding the role of PP2A-mediated phosphoregulation on responses of RAS-driven cancers to epigenetic therapies that has thus far been clinically disappointing. Notably, three different pharmacological PP2A-reactivating compounds resulted in similar effects on HDAC chromatin recruitment (Fig 4A–D), and DBK1154 synergized with HDACi in killing KRAS-mutant lung cancer cells (Fig S5B and C). Although DT061 was recently proposed to have toxic effects in cells by PP2A-independent mechanisms when used at high concentrations (Vit et al, 2022), the fact that all pharmacological PP2A activators, and genetic PP2A reactivation, phenocopy each other's effects (Fig 4A–D) provides strong evidence that their effects are mediated by PP2A activation. Collectively, these results encourage testing of the emerging PP2A-reactivating therapies for their impact on KRAS-mutant cancers in combination with epigenetic therapies.

Our data provide strong evidence that PP2A inhibition drives oncogenic transcription and globally impacts both DNA methylation and chromatin accessibility. The dominance of RNA expression regulation, as compared to DNA methylation or chromatin accessibility changes by PP2A inhibition (Fig S9B), is fully consistent with role of PP2A in the CDK9-mediated (RNAPII-driven) transcriptional elongation (Huang et al, 2020; Vervoort et al, 2021; Ohe et al, 2022). Indeed, both the phosphoproteome analyses used in this study and in other studies have recently demonstrated that PP2A regulates phosphorylation of several proteins directly involved in RNAPII complex function such as NELF-A, SPT5H, and RNAPII C-terminal tail phosphatase CTDSPL2 (Kauko et al, 2020; Vervoort et al, 2021; Ohe et al, 2022). However, the role of PP2A in DNA methylation and chromatin accessibility has been largely uncharacterized. Thereby, these data provide a rich recourse for future interrogation of roles of RAS and PP2A in regulation of these fundamental epigenetic mechanisms. As an example, although DNA methylation of the promoters is the most studied epigenetic mechanism regulating gene expression, recent reports indicate that cancer cells harbor hypomethylated regions at the intergenic regions (Lee & Wiemels, 2016). Thus, the observed hypomethylation at intergenic regions by PP2A inhibition indicates novel possible regulatory mechanisms for cancer progression. On the contrary, our results demonstrating

global chromatin opening by PP2A inhibition are consistent with recent results from mouse T cells in which deletion of regulatory B subunit of PP2A (PPP2R2D) resulted in chromatin opening (Pan et al, 2020).

Our results reveal dozens of RAS- and PP2A-regulated phosphorylation sites in epigenetic proteins previously implicated in transcription and cancer (Figs 2 and S2). Structurally, we provide evidence that at least some of these phosphosites are located on functionally important regions of the epigenetic proteins involved in oncogenic transcription (Fig 3). Mechanistically, we validate the impact of RAS and PP2A on selected phosphoprotein targets. Although the mechanism by which RAS and PP2A regulate CHD3 protein stability remains speculative, loss of CHD3 expression upon inhibition of RAS-elicited phosphorylation (Fig 3F) most likely contributes to the observed chromatin remodeling phenotype in RAS/PP2A-modulated cells (Fig 7). On the contrary, previous studies have shown that HDAC2 phosphorylation is required for its interactions with epigenetic multiprotein complexes such as Sin3, NuRD, or CoREST (Delcuve et al, 2012). In our data, both RAS and PP2A regulate C-terminal HDAC1 and HDAC2 phosphorylation on largely overlapping sites (Fig 2). Functionally, we validate that RAS and PP2A modulations regulate chromatin binding of HDAC1/2 and that this correlates with transcriptional activity of the highly HDAC-responsive SFRP1 promoter system. Naturally, both phenotypes can also be attributed to RAS/PP2A-mediated phosphorylation regulation of other NuRD complex components such as MTA2. In addition, we cannot exclude that RAS and PP2A regulate gene expression and chromatin accessibility by directly regulating histones or transcription factors (Gil & Vagnarelli, 2019). Interestingly, in addition to RAS/PP2A-mediated phosphoregulation of epigenetic proteins reported here, PP2A has been shown to dephosphorylate BRD4, HDAC 4/5/7, PRMT1/5, and TET2 that can contribute to chromatin structure regulation (Tinsley & Allen-Petersen, 2022). Therefore, these data collectively indicate that precise control of gene expression relies on the finely tuned balance between kinase and phosphatase activities.

Collectively, these data reveal a previously hidden layer of phosphoregulation of epigenetic gene regulation. Based on the results, it is very likely that convergence of the RAS and PP2A activities on the discovered epigenetic phosphoregulation contributes also to the synergism of RAS activation and PP2A inhibition in human oncogenesis. We further postulate that the discovered RAS/PP2A-mediated phosphorylation events are most probably relevant not only in cancer, but also in development and other diseases.

# Materials and Methods

### Phosphoproteome data filtering

The details of the phosphoproteomics pipeline and data analyses related to analyses of RAS and PP2A-regulated phosphosites are described in previous publications (Kauko et al, 2015, 2020). The raw data can be accessed via the PRIDE partner repository with the dataset identifiers PXD001374 (for RAS-regulated phosphosites) and PXD016102 (for PP2A-regulated phosphosites). For identification of

overlapping phosphosites regulated by RAS or any of the PP2A conditions, the following filtering criteria were used when assessing the phosphoproteome data normalized as described in Kauko et al (2015, 2020). For inhibition of phosphorylation by RAS targeting, fold change was −0.5 log$_2$ and FDR<0.1%. For PP2A-regulated proteins, fold change was 0.5 log$_2$ (for increased phosphorylation by PPP2R1A targeting) and −0.5 log$_2$ (for dephosphorylation by PME-1, CIP2A, and SET targeting) and FDR < 0.05%. The lower FDR criteria used for RAS data are due to the notion that in general, these earlier experiments (Kauko et al, 2015) had more variation between the replicate samples because of less sensitive mass spectrometry and inexperience in sample handling.

## Cell culture

KRAS-mutant lung cancer cell lines A549, H358, and H460, and HeLa cells were used. Furthermore, the normal bronchial epithelial cell line HBEC3-KT (HBEC), immortalized with CDK4 and hTERT, a kind gift of Prof. Jerry W Shay (Sato et al, 2013), was used. HCT116 SFRP1 promoter–GFP cells (Cui et al, 2014) were a kind gift of Prof. Stephen B Baylin. A549, H358, H460, and HBEC were authenticated (STR profiling) by the European Collection of Authenticated Cell Cultures in December 2018. The cells were cultured in medium conditions recommended by the providers for less than 4 mo before use in these experiments. All cells were regularly tested negative for mycoplasma.

## siRNA transfections and treatments

The siRNA transfections were done in 6-well to 96-well plates, using RNAiMAX transfection reagent according to the manufacturer's instructions, and siRNAs from QIAGEN and Eurofins.

### siRNA sequences and/or provider catalog #

siCtrl (#1): AllStars Neg. Control siRNA, Cat. No./ID: 1027281 (QIAGEN), siCtrl (#2): CGUACGCGGAAUACUUCGA (Eurofins), siPP2A-A: UUUUCCACUAGCUUCUUC A (Eurofins), siHRAS: GAACCCUCCUGAUGAGAGU (Eurofins), siKRAS: AGAGUGCCUUGACGAUACA (Eurofins), siNRAS: GAAAUACGCCAGUACCGAA, siPME (#1): GGAAGUGAGUCUAUAAGCA, siPME-1 (#2): UCAUAGAGGAAGAAG AAG A, siSET (#1): UGCAGACACUUGUGGAUGG (Eurofins), and siSET (#2): AAUGCA GUGCCUCUUCAUC (Eurofins). All siRNAs (CHD3, DNMT1, DOT1L, KDM1A, MLLT3, RNF168, and SMARCA4) for the cell viability assay were ordered from QIAGEN. AllStars Hs Cell Death siRNA, Cat. No./ID: 1027299 (QIAGEN), was used as a positive control (siCtrl +).

DNMT1 inhibitors (decitabine; AZA), BET inhibitors (iBET151, JQ1, mivebresib), HDAC inhibitors (panobinostat; TSA), KDM1A inhibitors (SP2509), and okadaic acid were used and purchased from SelleckChem. PP2A-reactivating compound DBK1154 was a kind gift of Dr. Michael Ohlmeyer (Atux Iskay; LCC).

## Nuclear fractionation

Cell fractionation was done using the Subcellular Protein Fractionation Kit for Cultured Cells from Thermo Fisher Scientific (#78840). Briefly, cells were harvested 48 h post-transfection using trypsin. One million cells were suspended in 100 $\mu$l of cytoplasmic extraction buffer and incubated for 10 min at 4°C with gentle mixing. Cells were then centrifuged at 500$g$ for 5 min, and the supernatant was collected as the cytoplasmic fraction in a new tube. Membrane extraction buffer (100 $\mu$l) was added to the pellets followed by vigorous vortexing for 5 s and incubation at 4°C for 10 min. The lysate was centrifuged at 5,000$g$ for 5 min, and the supernatant containing the membrane extracts was collected in a fresh tube. The pellet was suspended in the nuclear isolation buffer (50 $\mu$l) and vortexed for 15 s and further incubated for 30 min at 4°C with gentle rotation. Cells were centrifuged at 5,000$g$ for 5 min, and the supernatant containing the nuclear fractions was collected. The pellets were suspended in nuclear isolation buffer (50 $\mu$l) containing MNase (150 U) and 5 mM calcium chloride and vortexed for 15 s. The tubes were incubated at RT for 15 min to separate the chromatin-bound proteins. After vortexing again for 15 s, tubes were centrifuged at 16,000$g$ for 5 min and the supernatant containing the chromatin-bound proteins was collected in fresh tubes. Protein concentration of the fractions was determined using the BCA assay, and Western blotting was used to detect cellular localization of the desired proteins.

## Pull-down assays

Interaction between B56$\alpha$ and HDAC1 or between RNF168 and TP53BP1 was studied by co-immunoprecipitation analysis. The H460 cells were transfected with the respective plasmids using the jetPRIME transfection reagent. 48 h later, cells were harvested on ice by scraping and lysed in a buffer containing 100 mM NaCl, 1 mM MgCl$_2$, 10% glycerol, 0.2% protease inhibitor tablet (Roche), and 25 units/ml Benzonase (Millipore). Cells were rotated at 4°C on a roller, and 15 min later, the final concentration of NaCl and EDTA was increased to 200 and 2 mM, respectively. After further rotation of 10 min, cells were centrifuged at 21,000$g$ for 20 min. 10% of the lysate was stored as input, and the remaining was incubated with the 20 $\mu$l of prewashed GFP or FLAG-trap magnetic beads (ChromoTek GFP-Trap or Fab-Trap) at 4°C on a roller for 2 h. Postincubation, the beads were washed three times using the lysis buffer and eluted by adding 20 $\mu$l of 2× SDS loading buffer and boiling at 95°C for 10 min. Inputs and the CO-IP samples were further loaded on a 4–20% gradient gel to access the interactions.

## Western blots

Cells were lysed in RIPA buffer (50 mM Tris–HCl, pH 7.5, 0.5% DOC, 0.1% SDS, 1% NP-40, and 150 mM NaCl) with protease and phosphatase inhibitors (#4693159001 and #4906837001; Roche), followed by sonication at the highest setting with a pulse of ± 30 s. After centrifugation at 16,000$g$ for 30 min, lysates were collected in a fresh tube and protein concentration was determined using BCA assay (Pierce). 6× loading buffer was added to lysates, and they were boiled at 95°C for 10 min. Equal amounts of lysates were loaded on 4–20% precast gradient gels (Bio-Rad) and separated at 80–100 V. Proteins were blotted using PVDF membrane (Bio-Rad) and blocked for 1 h at RT. Membranes were incubated overnight with primary antibody followed by washing. For detection, HRP-labeled secondary antibodies (DAKO) followed by incubation with Pierce ECL Western Blotting Substrate (Thermo Fisher Scientific)

were used, or LI-COR Biosciences secondary antibodies (IRDye 680 or IRDye 800) were used followed by detection by Odyssey Imaging Systems or Bio-Rad Laboratories ChemiDoc Imaging Systems.

## Antibodies

The following antibodies, at the indicated dilutions, were used: FLAG (F3165, 1:1,000; Sigma-Aldrich); GAPDH (5G4-6C5, 1:5,000; HyTest); GFP (sc-9996, 1:500; Santa Cruz Biotechnology); HDAC1 (06-720-25UG, 1:1,000; Sigma-Aldrich) and HDAC2 (sc-9959, 1:1,000; Santa Cruz Biotechnology); H3 (sc-374669 (C-2), 1:1,000; Santa Cruz Biotechnology); PME-1 (sc-20086 (H-226), 1:1,000; Santa Cruz Biotechnology); and SET1 (I2PP2A (F-9), sc-133138, 1:1,000; Santa Cruz Biotechnology).

## Quantitative PCR

To determine the mRNA levels of the N-terminal FLAG-tagged RNF168, WT and mutant plasmids were transfected in cells. After 48 h, RNA was isolated using the NucleoSpin RNA kit (Macherey-Nagel), which was further reverse-transcribed to cDNA using the random primers, dNTP mix (Thermo Fisher Scientific), and cDNA kit Promega (M3681) M-MLV Reverse Transcriptase, RNase H Minus, Point Mutant as per the manufacturer's protocol. To specifically amplify the mRNA from the overexpressed RNF168, the forward primer targeting the N-terminal FLAG tag, while reverse the Exon1 of the RNF168, was used. The primer sequences are Forward (FLAG): 5'-ACGATGACGATAAAGCCGCCA-3' and Reverse (Exon1): 5'-AGGGA-CAGCATAAACTCGCCTT-3'.

The PCR amplification was done using the PowerUp SYBR Green Master Mix (Thermo Fisher Scientific) in QuantStudio 12K Flex Real-Time PCR System (Thermo Fisher Scientific). The gene expression levels were normalized to those of the housekeeping gene GAPDH, and the $2^{-\Delta\Delta CT}$ method was used to calculate the gene expression levels.

## Drug sensitivity assays

To determine the synergy between PP2A activation and HDAC inhibition, drug synergy screening was done. H460 cells were seeded in a 96-well plate (3,000 cells/well) and the next day treated with respective drugs for 48 h. Cell viability was measured using the CellTiter-Glo cell viability end-point assay (Promega), and the synergy was determined using the synergy finder tool (https://synergyfinder.fimm.fi/synergy/20220404143330175356/). HCT116 reporter cells were treated with the respective drugs at their IC50 concentrations, and 48 h later imaged for fluorescence using the IncuCyte ZOOM and/or S3 live-cell imaging, and then harvested using RIPA buffer. The fluorescence signal was analyzed using the ImageJ tool, whereas the GFP signal was determined using Western blotting.

## Anchorage-independent colony formation assay

For the anchorage-independent colony formation assay, which typically correlates with in vivo tumorigenicity, $2 \times 10^4$ cells were resuspended in 1.5 ml growth medium containing 0.4% agarose (4% Agarose Gel; Thermo Fisher Scientific/Gibco; top layer) and plated on 1 ml bottom layer containing growth medium and 1.2% agarose in a 12-well plate. After 14 d of growth, colonies were stained overnight with 1 mg/ml nitro blue tetrazolium chloride (Molecular Probes; NBT) in PBS. Colonies were imaged using a Zeiss SteREO Lumar V12 stereomicroscope. Analysis was done using ImageJ software. First, the background was subtracted using the rolling ball function with a radius of 23 µm, and then, auto-thresholding was applied to separate the colonies. Area percentage was calculated using the ImageJ built-in function "Analyze Particles" with exclusion of particles smaller than 200 µm$^2$ that are not considered colonies.

## RNA sequencing

HeLa cells were transfected with the siRNAs using Lipofectamine RNAiMAX. After 72 h, RNA, DNA, and protein were isolated using the AllPrep DNA/RNA/Protein Mini Kit (50) (QIAGEN, Cat. No./ID: 80004). RNA-seq was done at Finnish Functional Genomics Centre. First, the quality of the total RNA samples was ensured with Advanced Analytical Fragment Analyzer. Sample concentration was measured with Qubit Fluorometric Quantitation (Life Technologies). Library preparation was done according to Illumina TruSeq Stranded mRNA Sample Preparation Guide (part # 15031047). The first step in the workflow involves purifying the poly-A–containing mRNA molecules using poly-T oligo-attached magnetic beads. The samples were sequenced with Illumina HiSeq 3,000 instrument using single-end sequencing with 1 × 50 bp read length.

RNA-seq data quality was assessed with FastQC v0.11.7 ("FastQC," 2015; retrieved from https://qubeshub.org/resources/fastqc). Reads were trimmed with TrimGalore! v0.6.4._dev (https://zenodo.org/record/5127899) with the following parameters: –quality 20 –gzip -fastqc. Resulting trimmed single-end reads were aligned with STAR v2.5.3a with the following parameters: –quantMode TranscriptomeSAM –outSAMtype BAM SortedByCoordinate –chimOutType WithinBAM –twopassMode Basic –readFilesCommand zcat –genomeLoad NoSharedMemory –outReadsUnmapped FastX —bamRemoveDuplicatesType UniqueIdentical –seedSearchStartLmax 25 –outFilterMismatchNoverReadLmax 25 –outFilterMismatch-NoverReadLmax 0.04 –winAnchorMultimapNmax 100. Ensembl Homo sapiens GRCh38 v95 sequence and annotations were used as reference for the alignment. Gene counts produced with STAR were then assembled into a read count matrix within R v4.1.0 (R Core Team, 2019; retrieved from https://www.r-project.org/), and DESeq2 v1.34.0 (Love et al, 2014) was used to detect differentially expressed genes. For each comparison, prefiltering was applied to the corresponding data matrix: genes with zero read counts across all samples were removed, the lower quartile value of the resulting distribution was computed, and genes with overall read count lower than this value were removed. DESeq was run with default parameters. GO (Ashburner et al, 2000) term enrichment was computed with goseq v1.46.0 (Young et al, 2010) using non-electronic GO associations (IEA associations were removed).

## RRBS

DNA isolated from the same samples that were used for RNA isolation (using the AllPrep DNA/RNA/Protein Mini Kit) was used for RRBS. RRBS was done at Finnish Functional Genomics Centre.

Initially, the quality of the genomic DNA samples was ensured with Advanced Analytical Fragment Analyzer and concentrations were measured with Qubit Fluorometric Quantitation (Life Technologies). Library preparation was carried out according to the protocol adapted from Boyle et al (2012). Bisulfite conversion and sample purification were done according to Invitrogen MethylCode Bisulfite Conversion Kit. The samples were sequenced with Illumina HiSeq 3000 Instrument using paired-end sequencing with 2 × 50 bp read length.

RRBS data quality was assessed with FastQC v0.11.8 ("FastQC," 2015; retrieved from https://qubeshub.org/resources/fastqc). Reads were trimmed with TrimGalore! v0.6.4_dev (https://zenodo.org/record/5127899) (running on top of Cutadapt v2.7 [Martin, 2011]) with the following parameters: –quality 22 –phred33 –gzip –rrbs –fastqc –paired –cores 4. Resulting trimmed paired-end reads were aligned with Bismark v0.22.3 (Krueger & Andrews, 2011) using the Bowtie2 aligner v2.3.5.1 (Langmead et al, 2019) and the following additional parameters: –unmapped –ambiguous –ambig_bam –nucleotide_coverage –fastq. Methylation calls were extracted with the bismark_methylation_extractor and the following parameters: –paired-end –comprehensive –gzip –bedGraph –remove_spaces –buffer_size 80% –cytosine_report –ignore_r2 2. Data were aligned to the Ensembl Homo sapiens GRCh38 v95 genome, and corresponding annotations were used. Incomplete conversions were filtered out with the filter_non_conversion tools from Bismark. Finally, differential methylation analysis was run using Bioconductor (Huber et al, 2015) methylKit v1.12.0 (Akalin et al, 2012) packages running on R v3.6.1 (R Core Team, 2019; retrieved from https://www.r-project.org/). For the methylKit analysis, two scenarios were considered: CpG context and tiled window context. For single-base differential methylation calls, the following parameters were set: methylation difference 25%, adjusted q-value 0.01. On top of these, for the tiled analysis, window size and step size were set to 500 bp to generate tiling, non-overlapping windows. The whole analysis was bundled in a Snakemake pipeline, and software specifications were encapsulated in a Singularity container. The analysis was run on Snakemake v5.6.0 (Molder et al, 2021). Both the workflow and container recipes are available at https://github.com/ftabaro/MethylSnake

### ATAC sequencing

To profile the open chromatin regions, ATAC-seq was conducted, according to the protocol by Buenrostro and co-workers (Buenrostro et al, 2015). HeLa cells were transfected with siCtrl or siPP2A-A, and 72 h upon transfection, 50,000 cells were collected for analysis. At the end, the library sizes were determined by fragment analysis, and 2 × 75 paired-end sequencing was performed on NextSeq500 (Illumina) to yield an average of 50 M reads/sample. Sequencing library quality was assessed with FastQC v0.11.7. Reads were aligned on the human genome GRCh38 with Bowtie2 v2.3.4.1, converted to bam with samtools v1.8, and sorted with Picard SortSam v2.27.1. Bam files were indexed with Picard BuildBamIndex v2.27.1, and duplicated reads were marked with Picard MarkDuplicates v2.27.1. Peaks were called with MACS2 v2.1.0 with the following parameters: –gsize hs –qvalue 0.05 –format bam –nomodel –bdg –call-summits. Peak summits were annotated with

Homer v4.9 by running the annotatePeak.pl routine against hg38 annotation. Standardized peaks were computed starting from peak summit using the bedtools slop command v2.27.1 with parameters: -g $GENOME -l 250 -r 249. Read counts of standardized peaks were computed from bam files using the bedtools coverage command v2.27.1. The standardized peak read count was normalized using the median of ratio normalization and then converted to RPM. Finally, $\log_2$ fold change was computed from RPM values comparing signal intensities of corresponding genomic locations between siPP2A and siCtrl samples. Peaks with an absolute fold change greater than 0.5 were considered differentially accessible.

### Data integration

The significant gene list from the different omics data (q < 0.05) was integrated using the metascape tool (Zhou et al, 2019). The up-regulated genes from the RNA-sequencing data ($\log_2$fc > 1; q < 0.05), hypomethylated genes from RRBS data (methylation difference < 10%; q < 0.05), and DAPs with a fold change greater than 0.5 from the ATAC-seq data were used for the analysis. The tool identifies statistically enriched terms from all the datasets and performs hierarchical clustering based on $\kappa$-statistical similarities and a threshold $\kappa$ score of 0.3.

### Statistical analyses

Statistical analyses were completed using GraphPad Prism (GraphPad Software; www.graphpad.com). All experiments were repeated at least three times as indicated in the figure legends. The data were analyzed with the Mann–Whitney U test for significance (RNAi screens, reporter assays) or the two-tailed $t$ test (Western blotting). The statistical significance was elucidated as $P \geq 0.05$, not significant (ns), $P < 0.05$ (*), $P < 0.01$ (**), $P < 0.001$ (***), and $P < 0.0001$ (****).

## Data Availability

All omics data have been deposited in NCBI's Gene Expression Omnibus (Edgar et al, 2002) and are accessible through GEO Series accession number GSE220593. Other data are available from the corresponding author on reasonable request.

## Supplementary Information

## Acknowledgements

We thank the Proteomics, Screening, and Cell Imaging Core Facilities, and the Finnish Functional Genomics Center at Turku Bioscience Centre supported by the University of Turku, the Åbo Akademi University, and the Biocenter Finland. Taina Kalevo-Mattila is acknowledged for excellent technical help, and the entire Turku Bioscience Centre technical staff, for their

contributions. This work was supported, in part, by grants from the Academy of Finland (J Westermarck, 294850), Jane ja Aatos Erkko Foundation (J Westermarck), and InFLAMES Flagship Programme of the Academy of Finland (337530). M Sharma has been supported by the Turku Doctoral Programme of Molecular Medicine (TuDMM), Ida Montin Foundation, and the Paulo Foundation. We acknowledge the following colleagues for generously providing valuable research tools as listed in the Materials and Methods section: Prof. Krister Wennerberg, Prof. Jerry W Shay, Prof. Stephen B Baylin, and Dr. Michael Ohlmeyer. The Structural Genomics Consortium is a registered charity (no.: 1097737) that receives funds from Bayer AG, Boehringer Ingelheim, Bristol Myers Squibb, Genentech, Genome Canada through Ontario Genomics Institute [OGI-196], EU/EFPIA/OICR/McGill/KTH/Diamond Innovative Medicines Initiative 2 Joint Undertaking [EUbOPEN grant 875510], Janssen, Merck KGaA (aka EMD in Canada and US), Pfizer, and Takeda.

## Author Contributions

A Aakula: conceptualization, data curation, formal analysis, funding acquisition, validation, investigation, visualization, methodology, project administration, and writing—original draft, review, and editing.
M Sharma: conceptualization, data curation, formal analysis, funding acquisition, validation, investigation, visualization, project administration, and writing—original draft, review, and editing.
F Tabaro: data curation, formal analysis, visualization, and writing—review and editing.
R Nätkin: data curation, formal analysis, and visualization.
J Kamila: investigation.
H Honkanen: investigation.
M Schapira: resources, formal analysis, and visualization.
C Arrowsmith: resources and funding acquisition.
M Nykter: supervision, methodology, and writing—review and editing.
J Westermarck: conceptualization, data curation, supervision, funding acquisition, visualization, project administration, and writing—original draft, review, and editing.

## Conflict of Interest Statement

The authors declare that they have no conflict of interest.

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
