## [Reviewer comments · Life Science Alliance]

Life Science Alliance

RAS and PP2A activities converge on epigenetic gene regulation

Anna Aakula, Mukund Sharma, Francesco Tabaro, Reetta Nätkin, Jesse Kamila, Henrik Honkanen, Matthieu Schapira, Cheryl Arrowsmith, Matti Nykter, and Jukka Westermarck

DOI: <https://doi.org/10.26508/lsa.202301928>

Corresponding author(s): Jukka Westermarck, University of Turku

Review Timeline:	Submission Date:	2023-01-16
	Editorial Decision:	2023-01-31
	Revision Received:	2023-02-12
	Accepted:	2023-02-13

Transaction Report:

Please note that the manuscript was reviewed at Review Commons and these reports were taken into account in the decision-making process at Life Science Alliance.

Response to reviewers

We are extremely grateful for reviewer's enthusiasm over our data and their constructive suggestions for strengthening the manuscript. As listed below, we have been able to respond to all reviewer questions and provide significant new data clearly strengthening the main conclusions of the study. In total, the paper now contains 11 new data panels and 2 new supplementary tables based on the revision experiments.

Collectively, the study provides a novel explanation what are the cellular processes regulated by combined RAS activation and PP2A inhibition during human cell transformation. The data also opens new general understanding about regulation of epigenetic and gene regulation mechanisms by phosphorylation-dependent signaling. Thereby, in addition to specific data indicating role of some of the identified phosphorylation events in oncogenic gene expression regulation, the work provides a rich resource for future studies to understand how gene expression programs are regulated by phosphorylation in cancer and in other pathophysiological processes. We sincerely hope that the new stronger version of the manuscript can now be accepted for publication.

Reviewer #1 (Evidence, reproducibility and clarity (Required)):

I have only a few minor specific comments.

The overlap of PP2A and RAS regulated phosphoproteins in the gene ontology networks is made up of small numbers - 3/6 in term 0070087. When only 3 genes are in a category, given the reliability of GO terms, it doesn't generate much excitement.

Author's response: *We agree with this criticism but want to emphasize that rather than any individual category, the overall conclusion that RAS/PP2A signaling converge on epigenetic complexes and gene regulation was based enrichment of several related GO terms distributed over the clusters 2-4.*

Likewise the effect of knockdowns of putative targets in NSCLC cells was modest, with 10- 20% decrease in cell viability. I suspect many gene knockdowns might give a similar effect.

Author's response: *Although the small molecule inhibitors of these proteins showed more robust effects (Fig. S3C), we agree with this point and have modified the results text to more clearly indicate that these results were confirmatory and in line with known oncogenic role of the targeted proteins. We also removed the statement related to these results from the abstract.*

Line 299 starts a >1.5 pages long paragraph about CHD3 and HDAC1/2; it would be easier to read if this were two or three shorter paragraphs.

Author's response: *Due to additional results included in the revised manuscript, this data has now been divided to two figures (**New Fig. 3 and modified Fig. 4**) and consequently the text has been also split as suggested by the reviewer.*

The pulldown data (S5A) is done with over-expressed proteins and shows a weak interaction.

Without evidence for endogenous protein interaction, the conclusion that there is a substantial in vivo physiologic interaction between B56 α and HDAC1 must be qualified.

Author's response: *Unfortunately, even the best B56 α antibodies are cross-reactive with many cellular proteins and therefore based on general criteria they are unsuitable for pull-down experiments. On the other hand, it has been very challenging for us to detect endogenous B56 α from any antibody-based pull-downs as the relatively weak signal is always masked by the exact same size antibody fragments. Therefore, we simply had to rely on using GFP-trap approach with overexpressed HDAC1-GFP or B56-GFP proteins, but importantly in both cases detecting endogenous B56 α or HDAC1 from the pull-down samples (GFP-trap don't produce masking antibody fragments). We also want to emphasize that we do show evidence for interaction by reciprocal pull-downs that clearly increases the reliability of the observation (Fig. S3A).*

*To further substantiate the evidence that HDAC1 is a direct B56 α target protein, we now include schematic presentation of HDAC1 protein (**New Fig. 2C**) indicating location of conserved B56 α short-linear interaction motif (SLIM) adjacent to PP2A-regulated phosphorylation sites. The HDAC1 motif is highly like consensus B56 α binding motif (based on Kruse et al., Molecular Cell, 2018) and is located on the disordered protein region that is an additional requirement for B56 α targeting to the motif. We included in **New Fig. 2C** also another example of PP2A-targeted epigenetics proteins SMARCA4 with very well conserved B56 interaction motif. Existence of B56 recognition motifs in the presented epigenetics PP2A targets has been systematically analyzed in supplementary Table S3. Together, this evidence clearly supports the view that HDAC1, among the other emphasized epigenetics proteins, are PP2A-B56 target proteins.*

CROSS-CONSULTATION COMMENTS

I agree with reviewer 2 that there are shortcomings. If this is viewed as a resource, and not a strong conclusion paper, my feeling is that additional confirmation experiments would not add much. I agree they should be careful to discuss the limitations of the RNAi approach.

Author's response: *We have now discussed these limitations in the discussion part but want to point out that we already used chemical inhibitors and activators in the original version of the manuscript, and they did recapitulate the siRNA effects (Fig. 3F, 4C and 4H). We further want to emphasize that the underlying question addressed in this study was what are the cellular processes regulated by RAS activation and PP2A inhibition in relation to their co-operation during human cell transformation? The effects occurring during cellular transformation certainly are long-term effects. Thereby use of siRNA to study conditions related to cellular transformation was a conscious decision to better model the addressed conditions over using short term chemical treatments. We even established the stepwise transformation model to confirm that the long-term effects in HDAC regulation are truly seen in conditions in which RAS activation and PP2A inhibition co-operate in human cell transformation (Fig. 4G,H and S5B,C). Thereby the data do collectively demonstrate that in conditions modelling human cell transformation, RAS and PP2A activities converge on epigenetic proteins and on transcription (i.e. the main conclusion of the study). This is now better emphasized in the text (ln. 369) "These results demonstrate that combined requirements for human cell transformation that is RAS activation and PP2A inhibition, results in enhanced HDAC1/2 recruitment to chromatin, and that this can be reverted by pharmacological PP2A reactivation"*

Reviewer #2 (Evidence, reproducibility and clarity (Required)):

Major comments:

1. The investigation and characterization of the phosphosites that are common to both RAS and PP2A is an important question, as stated by the authors. However, the authors hardly investigated the potential roles of these common phosphosites (only CHD3 S713 has been partially investigated) but rather relied on knockdown by siRNAs of the factors, which limits the conclusions of the manuscript as it remains unknown whether these phosphosites have any effect on protein activity and/or interactions.

Author's response: *In response to reviewer's comment we now provide evidence that alanine mutation of PP2A and RAS-regulated phosphorylation site S481 on RNF168 results in increased binding of RNF168 to its co-factor TP53BP1 (New Fig. 3C,D). Functionally dephosphorylation mimicking alanine mutant was impaired in transcriptional activation as compared to either wild-type RNF168, or S481D mutant mimicking the constitutively phosphorylated RNF481 (New Fig. 4I,J). Whether these two RNF168 S481 phosphorylation-dependent functions are functionally related remains as a question to be addressed in the future but both experiments do provide requested evidence that the identified phosphosites have effect on protein activity and/or interactions.*

2. The major technical limitation of the manuscript is the dependence on siRNAs to investigate RAS and PP2A. Knockdown by siRNAs takes a long time, which limits the conclusions that can be drawn as the results are going to be a mixture of direct (loss of RAS/PP2A) and indirect (cellular responses to the direct effects) effects. Typically, changes in gene expression, DNA methylation, and chromatin accessibility could be explained, at least in part, by indirect effects of the knockdown (changes in cell cycle, cellular responses to stress induced by the knockdown...). I think it will be important to confirm on some target genes that the main results of the manuscript are direct effects by using known small molecule inhibitors with short treatment time.

Author's response: *We have now discussed these limitations in the discussion part. We want to point out that we already used chemical inhibitors and activators in the original version of the manuscript, and they did recapitulate the siRNA effects (Fig. 3F, 4C and 4H). Further, the underlying question addressed in this study was what are the cellular processes regulated by RAS activation and PP2A inhibition in relation to their co-operation during human cell transformation? The effects occurring during cellular transformation certainly are long-term effects. Thereby use of siRNA to study conditions related to cellular transformation was a conscious decision to better model the addressed conditions over using short term chemical treatments. We even established the stepwise transformation model to confirm that the long-term effects in HDAC regulation are truly seen in conditions in which RAS activation and PP2A inhibition co-operate in human cell transformation (Fig. 4G,H and S5B,C). Thereby the data do collectively demonstrate that in conditions modelling human cell transformation, RAS and PP2A activities converge on epigenetic proteins and on transcription (i.e. the main conclusion of the study). This is now better emphasized in the text (ln. 369) "These results demonstrate that combined requirements for human cell transformation, that is RAS activation and PP2A inhibition, results in enhanced HDAC1/2 recruitment to chromatin, and that this can be reverted by pharmacological PP2A reactivation"*

3. The genome-wide data do not seem to have been submitted to the GEO (or I could not find the information), which also means that it is not clear how many biological replicates have been performed.

Author's response: Thank you for notifying us on this important deficiency. All OMICs data have now been submitted to GEO and are accessible through accession number GSE220593. The data will be made fully available without restrictions at acceptance. Meanwhile, token 'ahadcwkkfxoblij' allows anonymous, read-only access to GEO record GSE220593 while the record remain private before publication. All OMICs data is based on three biological replicates/condition.

4. Generally, the authors should put more information in the Legends/Methods as several key information are missing (see Minor Comments).

Author's response: We agree with this criticism and have supplemented both sections with more detailed information.

5. The authors should integrate more their RNA-seq, RRBS, and ATC-seq data as these datasets have been generated in the same cell line (I suppose RRBS is also in HeLa, see Minor Comment 2). Do the authors see consistent changes on RRBS/ATAC-seq for the upregulated/downregulated genes?

Author's response: In response to this request, we now provide data integration as new publication and supplementary figures. Indeed, there was a strong overlap in enriched biological processes based on PP2A regulated ATAC-seq. and RNA-seq. data (**New Fig. 7F**), as well as between PP2A regulated RRBS and RNA-seq. data (**New Fig. S9**). We also integrated all three levels of regulation. Naturally the overlap across three technologies was more limited, but interestingly cancer and RAS/PP2A relevant processes such as "Signaling by RTK" and "Kinase signaling" (**New Fig. S9**). The transcription changes are clearly dominating over the RRBS and ATAC-seq. changes and the mechanistic basis of that is now discussed starting from ln. 665. Importantly, based on the data PP2A inhibition results in a) Increased transcription, b) DNA hypomethylation and c) preferentially open chromatin. As now discussed in the paper, all these changes enhance (oncogenic) transcription and thus demonstrate concerted action by PP2A inhibition at the level of epigenetic gene regulation (ln. 585). To better emphasize this, we fused the previous figure 6 and 7 as **New Fig. 7** in which both the RRBS and ATAC-seq. data is shown. The rest of the figure panels have been moved to **Modified Fig. S7 and New Fig. S8**.

Minor comments:

1. Did the authors performed a total (with rRNA depletion) or a poly(A)+ RNA-seq?

Author's response: It was poly(A)+ RNA-seq and this has now been mentioned in materials and methods.

2. In the Methods section for the RRBS, it is written that the DNA was isolated from the same samples. Is it the same samples as the RNA-seq? More precision is required.

Author's response: Yes, the DNA & RNA comes from the same samples isolated using the AllPrep DNA/RNA/Protein Mini Kit (50) Qiagen Cat. No. / ID: 80004. This is now indicated in the materials and methods.

3. It would also be useful to put in the legends the cell line used in each experiment.

Author's response: This information is provided now either in the figure legends of the main figures or in supplementary figures

4. Figure 3, Figure 4, and Figure S5: I could not find any information on the treatment time and the concentrations of the small molecule inhibitors used. These information need to be added to the legends.

Author's response: We apologize for these deficiencies which have now been corrected in corresponding figure legends.

5. Figure 3B: the authors need to performed qRT-PCR to show that the overexpression is similar between the different conditions. Right now, the differences could be explained by a difference in transcription between the different constructs.

Author's response: We attempted to also address this minor comment, but for unclear reason could not get the PCR working for CHD3 although other targets in the same assay, such as RNF168 (**New Fig. S3D**) worked fine. The opposite effects of CHD3 A -and D-mutants are fully in line with the model of phosphorylation-dependent protein stability effects. On the other hand, it is rather unlikely that one nucleotide mutation in the CHD3 cDNA coding region would result in decreased transcription from one construct (A), but increased transcription from another (D).

6. Also, do the mutations affect CHD3 chromatin association or interaction with other NuRD components? This kind of straightforward experiments would clearly improve the interest of the manuscript as it will provide information on the potential roles of phosphosites.

Author's response: Due to marked differences in protein stability of CHD3 mutants, it would have been challenging to get conclusive results in interaction studies as expression levels themselves naturally would have affected the results. Therefore, we rather focused on the newly generated RNF168 mutants as described in our response to reviewer's question 1 above. We believe that the data shown in **New Fig. 3** now provides the requested evidence that the identified phosphosites have effect on protein activity and/or interactions. This has now been mentioned in the manuscript ln. 308 "Collectively, these data provide important indications about functional relevance of RAS -and PP2A co-regulated phosphosites in epigenetic proteins. However, understanding of the functional role of each identified phosphorylation site reported here will require extensive validation experiments outside the scope of this resource article"

7. Figure 3C, E, G, and I: A nuclear loading control is required for each experiment. Also, western blots on whole cell extracts are required to see if the changes in nuclear/chromatin level are not just explained by a change in the total expression of HDAC1 and HDAC2 following siRNA treatment.

Author's response: As requested, we now provide new data demonstrating that none of the siRNA or chemical manipulations affect expression of HDAC1 or HDAC2 in whole cell extracts (**New data S5F**). We have also modified our conclusions so that we only mention increased chromatin loading of HDAC upon PP2A activation and RAS inhibition, but do not claim anything about nucleoplasmic retention of these proteins.

8. Lines 552-555: I am not convinced that the presence of DOT1L among the regulators associated with open promoter regions provides a direct link between the phosphoproteome and ATAC-seq data. DOT1L is a methyltransferase associated with transcription initiation and transcription elongation and therefore it is not surprising to find this protein in open promoter regions. In addition, to claim a direct link would require data showing that protein phosphorylation of DOT1L regulates its recruitment to promoter regions.

Author's response: We agree with this comment and have removed any discussion related to the matter.

9. Figure 7F/G: Are the overlaps significantly enriched?

Author's response: In response to this request we performed statistical analysis which revealed that the overlap of PP2A regulated DAPs with the open chromatin areas in clinical lung cancer samples was significantly enriched in specific chromosomes (chromosomes 6, 7, 18, 19

and 22)(**New Table S4**). We further analyzed the nearest genes associated with overlapping DAPs between PP2A inhibited samples and lung cancer samples and identified several cancer relevant genes especially from the lung cancer relevant chromosome 19 (**New Table S5**). These new results have been described in ln. 573-583. Together these results provide first evidence that PP2A inhibition regulates chromatin accessibility and provide indications for potential cancer relevance for this newly identified epigenetic function for PP2A.

CROSS-CONSULTATION COMMENTS

If the manuscript is clearly presented as a resource paper, I agree with reviewer 1. My major comments 1 and 2 (knockdown of total proteins rather than looking at phosphoresidues, RNAi) can be addressed in the discussion rather than experimentally.

Author's response: We truly appreciate both reviewer's very positive view on the manuscript and that they see its value as an important resource for future studies. Further, although the reviewer 2 indicated that his comments could have been addressed also in the discussion, we felt answering to them also experimentally important for increasing the value of the manuscript for the field.

January 31, 2023

RE: Life Science Alliance Manuscript #LSA-2023-01928-T

Prof. Jukka Westermarck
University of Turku
Finland

Dear Dr. Westermarck,

Thank you for submitting your revised manuscript entitled "RAS and PP2A activities converge on epigenetic gene regulation". We would be happy to publish your paper in Life Science Alliance pending final revisions necessary to meet our formatting guidelines.

- please address Reviewer 1's remaining comments
- please add ORCID ID for corresponding author-you should have received instructions on how to do so
- please upload both your main and supplementary figures as single files and add a separate figure legend section (with both main and supplementary figure legends) to the main manuscript text
- please add a Running Title, an alternate abstract, and a category in our system
- please add the Twitter handle of your host institute/organization as well as your own or/and one of the authors in our system
- please consult our manuscript preparation guidelines <https://www.life-science-alliance.org/manuscript-prep> and make sure your manuscript sections are in the correct order
- please add the Author contributions to the main manuscript text
- please use the [10 author names, et al.] format in your references (i.e. limit the author names to the first 10)
- GEO dataset GSE220593 should be made publicly accessible at this point

Figure Check:

- please make sure sizes are indicated next to all blots
- please add a scale bar to Figure S5

A. FINAL FILES:

B. MANUSCRIPT ORGANIZATION AND FORMATTING:

Sincerely,

Reviewer #1 (Comments to the Authors (Required)):

The authors have adequately addressed all my points.

I have only two small text comments:

1. Line 684: I do not think there is any RNA polymerase II CTD phosphatase called CTSL2. The closest one I can think of is CTDSP2 (there is also CTDSPL2).
2. I think the authors should also say that some questions have been raised on the specificity of PP2A activators (Vit, G. et al, Chemogenetic profiling reveals PP2A-independent cytotoxicity of proposed PP2A activators iHAP1 and DT-061, EMBO Journal, 2022).

Reviewer #2 (Comments to the Authors (Required)):

The authors have fully addressed my concerns.

February 13, 2023

RE: Life Science Alliance Manuscript #LSA-2023-01928-TR

Prof. Jukka Westermarck
University of Turku
Finland

Dear Dr. Westermarck,

Thank you for submitting your Research Article entitled "RAS and PP2A activities converge on epigenetic gene regulation". It is a pleasure to let you know that your manuscript is now accepted for publication in Life Science Alliance. Congratulations on this interesting work.

DISTRIBUTION OF MATERIALS:

Again, congratulations on a very nice paper. I hope you found the review process to be constructive and are pleased with how the manuscript was handled editorially. We look forward to future exciting submissions from your lab.

Sincerely,
